# Management of Slum-Based Urban Farming and Economic Empowerment of the Community of Makassar City, South Sulawesi, Indonesia

**Batara Surya [1,*], Syafri Syafri [2], Hadijah Hadijah [3], Baharuddin Baharuddin [4], Andi Tenri Fitriyah [5] and Harry Hardian Sakti [6]**

1   Department of Urban and Regional Planning, Faculty of Engineering, University Bosowa Makassar, Makassar City 90231, Indonesia
2   Department of Urban Ecology, Faculty of Engineering, University Bosowa Makassar, Makassar City 90231, Indonesia; syafri@universitasbosowa.ac.id
3   Department of Aquaculture, Faculty of Agriculture, University Bosowa Makassar, Makassar City 90231, Indonesia; hadijah.mahyuddin@universitasbosowa.ac.id
4   Department of Agricultural Socio-Economic, Faculty of Agriculture, University Bosowa Makassar, Makassar City 90231, Indonesia; baharuddin@universitasbosowa.ac.id
5   Department of Agricultural Agribusiness, Faculty of Agriculture, University Bosowa Makassar, Makassar City 90231, Indonesia; tenri.fitriyah@universitasbosowa.ac.id
6   Department of Urban and Regional Planning, Faculty of Engineering, University Muhammadiyah Bulukumba, South Sulawesi 92511, Indonesia; revplano07@gmail.com
*   Correspondence: batara.surya@universitasbosowa.ac.id

**Abstract:** The handling of slums based on urban farming and community empowerment is oriented toward improving the welfare and independence of the community. This study aimed to (1) analyze the influence of business motivation, human resource capacity, community participation, and economic business management on the economic empowerment in the handling of slums; (2) analyze the direct and indirect effects of urban farming, community capacity, and capital support efforts to improve the welfare and independence of the community; and (3) formulate a sustainability model for community-based slum management in the Metro Tanjung Bunga area. The research approach used was a sequential explanatory design, which is a combination of quantitative and qualitative research, and data were obtained through observation, surveys, in-depth interviews, and documentation. The research findings show that business motivation, human resource capacity, community participation, and economic business management have a significant effect on economic empowerment. The direct influence of urban farming increases the welfare of the community by 27.66%, strengthens the capacity of the community to increase welfare by 55.95%, and provides business capital support to improve community welfare by 36.72%. Urban farming, community capacity-building, and business capital support provide a positive effect on improving the welfare and independence of the community in slums. Sustainability in slum settlements includes infrastructure, developing economic potential, and community participation.

**Keywords:** slum; urban farming; economic empowerment; economic productivity; people's autonomy

## 1. Introduction

Industrialization and modernization are the triggering factors that push the dynamics of urban growth toward over-urbanization and maximum compaction. With the continuous acceleration of economic globalization and global urbanization, urban systems are undergoing an important

transformation and reconstruction [1]. Urbanization in the case of large and metropolitan cities is characterized by an increase in population, changes in land use, and complexity of urban ecosystems. Urban areas are significantly different compared to areas located outside cities when describing ecological problems, including the quality of soil cover. Most of the land in urban areas has decreased environmental quality due to human activities [2]. Furthermore, contemporary urbanization differs from historical patterns of urban growth in terms of scale, rate, location, form, and function [3]. The metropolitan-based urbanization process involves the development of peri-urban areas, which could be defined as transitional zones between urban and rural areas characterized by integrated mixed structures of agricultural and non-agricultural activities [4,5].

In the figures of the current total world population, it is recorded that approximately 50%, or 4.2 billion people, live in urban areas. This figure is projected to continue to increase to 70%, i.e., an additional 2.5 billion people who will live in urban areas [6]. Furthermore, in line with the increase in the total population, the likelihood of an increase in the size of the poor population in urban areas will also rise, and if not managed properly, urbanization will actually have a bad impact on the economy of a country, especially developing countries, including Indonesia. Thus, excessive urbanization and maximum compaction will create new slums, congestion, and urban flooding, and will widen socio-economic inequality. A major challenge for sustainable urbanization policies and strategies is how to address the complexity of urbanization, especially the ongoing growth of informal settlements and slums in developing countries [7,8]. Thus, urbanization is synonymous with the development of big cities in developing countries, the existence of poor communities and slums [9].

The World Bank projects that 220 million Indonesians will live in cities and towns by 2045. Furthermore, urbanization in Indonesia is predicted to continue to increase from 56% to 70% of the current total population. Every 1% increase in urbanization in the Asia Pacific and East region can, on average, increase per capita by 2.7%. However, for every 1% increase in urbanization in Indonesia, it is only possible to leverage 1.4% per capita. This means that the contribution of urbanization in Indonesia is still considered quite low when compared to other developing countries in East Asia and the Pacific [6]. The rate of urbanization in Indonesia today is extraordinary in terms of the rapid growth rate of the city [10,11]. Thus, urbanization in Indonesia still requires serious handling and attention from the government [12].

The dynamics of the development of metropolitan cities in developing countries, including in Indonesia, are positively associated with socio-economic differences regarding the fragmentation of people's lives in response to changes in environmental stimuli. Ongoing urbanization requires a much faster policy response at all levels, from local to global [13]. In the process of urbanization in Indonesia, the spatial form of the main cities is changing into very complex industrial spatial usage. In other words, the spatial form is being transformed from scattered small industrial land pieces interspersed with other types of urban land within the urban core to a polycentric pattern of large patches with greater distances between patches [14,15].

The rapid and revolutionary growth of the metropolitan city is a consequence of changes in spatial planning and physical land use and the result of differences in economic interests and their impact on community segmentation, environmental degradation, poverty, and slum settlements. Rapidly developing urban populations in cities that live in poverty in informal settlements pose major problems for urban health, safety, and risk reduction [16,17]. Poverty and its negative consequences limit human development, so that the poor are often at the ultimate level of vulnerability to health and economic turmoil and natural disasters [18].

The growth of the urban density in Indonesia during the period of 2014–2019 increased by 2.75%. Such a percentage is higher than the pace of the national population growth of approximately 1.17%. Furthermore, from the prediction of the population in 2045, it was calculated that 82.37% of the population would live in urban areas, and only 20% of people would live in suburban areas [19]. Insufficiently planned urbanization tends to affect urban issues, such as pollution, slum growth, clean water services, fulfilment of food needs, and electricity networks. In the long term, this will

affect the gap between districts and cities, as well as between villages and cities. Urban slums in developing countries are nothing but the result of informal, illegal, and unplanned urban growth [20]. Slum development is fueled by a combination of rapid rural-to-urban migration, spiraling urban poverty, the inability of the urban poor to access affordable land for housing, and insecure land tenure [21]. The urban population in Indonesia continues to grow and the ability of people to access economic resources is affected, leading to poverty, slum growth, and a decrease in the quality of the environment. The growth of slums is not a wholly organic process that occurs within the domestic conditions of a country; rather, it is one of the results of globalized and neoliberal capitalism [22]. Urbanization includes four subsystems, namely, economic urbanization, population urbanization, social urbanization, and land urbanization [23].

The modernization of the development of Makassar City as the center of the development of Eastern Indonesia and the national strategic area is an attractive factor for the suburban population and the surrounding regions. Urbanization is an inevitable trend of modernization and development, and it is a fundamental transformation of economic structure, social structure, production, and lifestyle, as well as a gradual process of long-term accumulation and development [24]. The slums of Makassar City are predominantly located in the riverbank, urban functional, and coastal areas. These locations are potential flood areas and have bad sanitation and a highly polluted environment. To protect slums in such locations is a serious problem to be overcome through the developmental policies of the city government. The upgrading of slums requires synergistic behavior between community and institutional actors [25]. Thus, government transparency, accountability, and responsiveness are needed for citizens, and the primary focus needs to be on the ways in which citizen-led accountability strategies can work to improve services for the poor and marginalized [26]. The distribution of the slums of Makassar City, based on its typology, can be categorized into three main groups: (1) Lowland slum, with a width of 129.71 hectares; (2) slums along the water's edge, with an area of 50.26 hectares; and (3) slums on water, with a width of 29.76 hectares. Such different typology is influenced by the conditions of the location, the community's work orientation, and the status of land ownership.

The number of places to live within slums, characterized by a lack of sanitation and public services, inadequate construction conditions, and irregular land ownership, also grows at a very high rate [27]. In addition, development such as that of Makassar City, which predominantly tends to lean on economic growth, causes a sharp rise in the demand for housing, which results in the development of slums by the poor who are unable to fulfil their housing needs independently. The large-scale emergence of slums can be seen as a side effect of massive global urbanization processes that have reached a new level of intensity during the past 40 years, during which time 20 new cities reached the status of mega cities [28].

The position of Makassar City as the center of national activities (PKN) and the center of the development of Eastern Indonesia is a driving force for excessive urbanization and maximum compaction. The rapid growth dynamics of Makassar City have resulted in a very significant increase in population and socio-economic problems at the micro level of the community. Excessive urbanization and maximum compaction in the dynamics of the development of Makassar City have an impact on the use of very complex spaces [29]. The population of Makassar City was 1,408,072 people in 2014, which increased by 1,526,677 people in 2019. This figure shows that during the five-year period, the population growth rate of Makassar City increased by 1.41% and exceeded the average national population growth of 0.7%.

The poverty in the population of Makassar City continues to increase in line with the above-mentioned increase in population size, that is, by 30,401 people, resulting in the existence of 729 hectares of slums. Furthermore, the poverty rates and size of slums also tend to increase, which has an impact on the complexity of ecosystems and changes in land use in locations that must be protected, including riverbanks and coastal areas. The data obtained show that in Makassar City, a total of 103 locations were recorded to house slums, increasing to 127 in 2019, which were distributed across 15 sub-districts. This figure shows that the slums of Makassar City experienced the addition of 24 new

locations [30]. This fact illustrates that the developing slum settlements in Makassar City are influenced by several factors, namely excess urbanization, poverty, and the powerlessness of the community to access the city's economic resources. Informal settlements i.e., slums emerge from the interplay of multidimensional factors related to urbanization and sustainability [31].

Community economic empowerment is an effort made to increase the community's capacity to generate added economic value. The enhancement of community capacity is carried out in four ways, namely (i) access to resources, (ii) access to technology, (iii) access to markets, and (iv) access to demand. The concept of economic empowerment was born as the antithesis of the urban development and industrialization model that did not side with, the majority of the people. Thus, there is a significant relationship between income inequality and economic growth [32]. Furthermore, income inequality can be overcome through community economic empowerment which is carried out comprehensively and sustainable [33].

The concept of economic empowerment basically refers to four main things, among others: (1) The process of centralizing power is built from the centralization of control over production factors; (2) The concentration of power in the factors of production will give birth to a working society and a society with marginal entrepreneurs; (3) power will build superstructure or knowledge systems, political systems, legal systems, and manipulative ideologies to strengthen and legitimize; and (4) co-optation of knowledge systems, legal systems, political systems, and ideologies, which will systematically create two groups of people, namely the empowered and the disabled. These four things will have an impact on the dichotomy, namely ruling society, and ruling humans. In order, to free the situation of control and control, it must be done through the empowerment process for those under control (empowerment of the powerless). Thus, community economic empowerment based on urban farming will encourage increased productivity of economic enterprises towards the sustainability of the community economy. Five sets of action principles that are ideal for promoting community productivity, namely: (1) Empowerment and integration of underprivileged communities; (2) Promotion of environmentally friendly agricultural practices; (3) Protection of nature, resources and landscape culture; (4) Support for the community local, and (5) Education for sustainable development [34].

Slum growth in the Metro Tanjung Bunga area of Makassar City is positively related to high unemployment, poverty, and socio-economic inequality in the community. The results of field research show that the community's low access to economic activities and its inability to adapt to changes in environmental stimuli have an impact on marginalization and poverty. Furthermore, the determinant factors of the development of slums and community poverty will require empowerment support to encourage increased economic productivity of the community. Urban farming in the context of economic empowerment in this study is oriented toward increasing the productivity of community businesses in a bid to increase the sustainability of slum management. Thus, the aim of this study is to answer three research questions: (1) What influence do business motivation, human resource capacity, community participation, and economic business management have on urban farming-based economic empowerment in the handling of slum settlements in the Metro Tanjung Bunga area of Makassar City? (2) What are the direct and indirect relationships of urban farming, community capacity, and business capital support for improving the welfare and independence of the community in the Metro Tanjung Bunga area of Makassar City? (3) What is the sustainability model for handling slums in the Metro Tanjung Bunga area of Makassar City?

## 2. Conceptual Framework

The handling of slums based on their residents' economic empowerment is needed to support the reinforcement of the ownership of the factors of production, distribution, and marketing to improve welfare, as well as community institution capacity reinforcement to obtain information, knowledge, and skills. Community-based economic empowerment can increase economic income and people's insight into becoming more open to the economic community, as well as the newly formed

community [35]. The urban farming approach is basically oriented toward efforts to improve the productivity of the economic endeavors of the community using the limited land available to provide ecological benefits for the environment. Urban trails and green spaces have become ubiquitous features in cities, in part due to their ability to provide ecological benefits to the built-up environment [36].

Urban farming is not only focused on the empowerment of the urban poor community in slum areas, but it is also expected to encourage improvement of community welfare through the marketing system maintained by the community itself. Furthermore, urban farming is applied within a neighborhood environment for strengthening the sense of togetherness to create a culture of helping one another within an urban community. The practice of urban agriculture has gained importance due to the rising rate of the urban population and the subsequent rise in poverty in developing regions [37]. The conceptual framework of this study is outlined Figure 1.

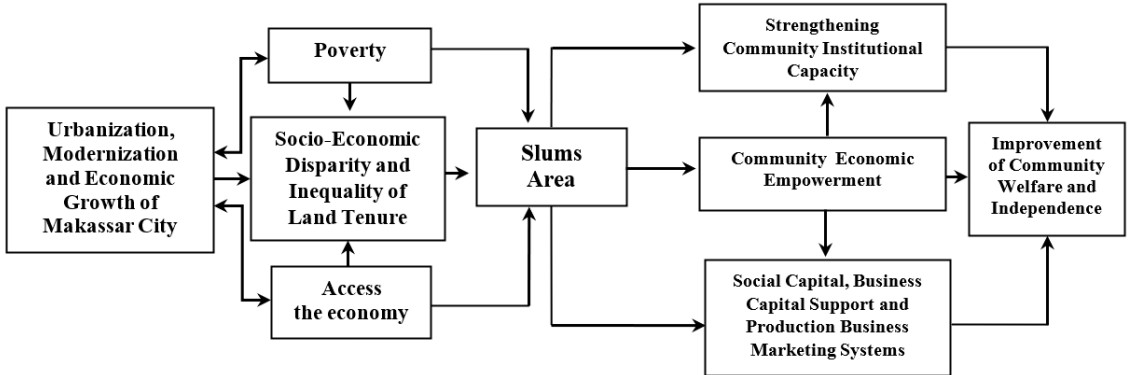

**Figure 1.** Conceptual Framework for Managing Slum-Based Urban Settlements and Community Economic Empowerment.

## 2.1. Urban Farming Based on Community Economic Empowerment

Urban farming is a concept implemented by utilizing limited urban land such as living quarters (balconies, roofs, or yards), roadsides, and even riversides to become productive gardening places. The urban farming system, which is a cultivation practice, from planting seeds to harvesting, distributing produce, and marketing is done in the city itself which is accessed by the community. Urban farming has a big impact on the survival of urban communities. The concept of urban farming basically offers a solution by creating green open land in the middle of dense urban buildings. The implementation of urban farming in urban development will function in controlling pollution towards a comfortable, healthy environment and increasing aesthetic value. Urban planners and agricultural leaders have argued that cities will need to produce food internally to respond to demand by increasing population and to avoid paralyzing congestion, harmful pollution, and unaffordable food prices [38].

Planting systems with urban farming patterns such as verticulture, hydroponics and aquaponics can easily be implemented by the community to be applied in a limited area. Urban agriculture manifests in a variety of different forms, often with different functions [39]. Furthermore, the role of city governments is very important in providing special regulations to support the implementation of urban farming, in relation to land use policies. Thus, urban farming based on community economic empowerment will have an impact on increasing the productivity of economic enterprises, reducing unemployment, and poverty for people located in slum areas. Urban food is mainly produced at managed agro-ecosystems [40]. In addition, forests, (industrial) rooftop gardens, residential and community gardens, containers on balconies, vacant land, edible landscaping, vertical edible green infrastructure as well as marine and freshwater systems are spaces for urban food products [41,42].

Empowerment refers to the ability of people, especially vulnerable and vulnerable groups, to have access to productive sources that enable them to increase income and obtain the goods and services they need and participate in the development process. Community economic empowerment is

defined as an effort to build people's power in the economy in particular by encouraging, motivating, and exploring their potential so that conditions will change from helpless to powerless with the manifestation of real actions to increase dignity from the economic side and escape. from poverty and underdevelopment. Economic empowerment is carried out in order to improve the standard of living of the community, so it is necessary to have an empowerment pattern that is right on target to provide opportunities for the poor to plan and implement predetermined development programs. Furthermore, microcredit on increasing participation in the overall decision-making process, in legal awareness, independent movements, and mobility, as well as enhancing living standards to encourage sustainability empowerment [43].

*2.2. Economic Business Sustainability and Community Independence*

The sustainability of economic enterprises is closely related to efforts to increase community productivity. Furthermore, community economic empowerment can be carried out by strengthening control of distribution and marketing, strengthening to get adequate wages/wages, and strengthening in obtaining information, knowledge and skills to increase the community's ability to be able to stand on their own. Economic empowerment can produce prosperity, where welfare is the dream of every person and every country [44]. Furthermore, community economic empowerment is a process in which the community, especially those who are resource poor, women, and other neglected groups are supported so that they are able, to improve their welfare independently [45].

Strategies that can be carried out in community economic empowerment include: (1). Motivation. Every household needs to be encouraged to form groups which are the institutional mechanism for organizing and implementing community development activities. The group is further motivated to be involved in income generation activities using their own resources and capabilities; (2). Awareness raising and capacity training. Increasing public awareness can be achieved through basic education, improved health, immunization, and sanitation. Meanwhile, vocational skills can be developed through a participative way; (3). Self, management. Each community group must be able to elect its own leader and organize its own activities, such as holding meetings, conducting recording and reporting, operating savings and credit, conflict resolution and community ownership management; (4). Resource mobilization. Mobilizing community resources requires developing methods for pooling individual resources through regular savings and voluntary contributions with the aim of creating social capital; (5). Network Development and Development. Organizing a non-governmental organization must be accompanied by an increase in the ability of its members to build and maintain networks with various social systems around them. This network is very important in providing and developing various access to resources and opportunities for increasing the empowerment of the poor [46].

Several forms of community economic empowerment practices that can be implemented include: First, providing capital assistance. Community empowerment efforts in the economic sector through the capital aspect are carried out by providing capital assistance with the aim of not causing community dependence. Solving this aspect of capital is done through the creation of a new conducive system for micro, small and medium enterprises to gain access to financial institutions. Second, infrastructure development assistance. The availability of marketing and/or transportation infrastructure from the production location to the market will reduce the marketing chain and will ultimately increase the income of micro, small and medium farmers, and entrepreneurs. Third, assistance. The main task of mentoring is to facilitate the learning process or reflection and become a mediator for strengthening partnerships between micro, small and medium enterprises, and large businesses.

Fourth, strengthening institutional capacity. The individual approach did not produce satisfactory results. Therefore, the approach is carried out with a group approach. The reason is that capital accumulation will be difficult to achieve among the poor, therefore capital accumulation must be carried out together in a group or joint venture. Likewise, with the distribution problem, it is impossible for the poor to control the distribution of production outputs and production inputs individually, so that through groups they can build strength to determine distribution. Fifth, strengthening business

partnerships. High competitiveness only exists if there is a link between large and medium and small ones. Because it is only with fair production linkages, efficiency will be built. Through partnerships both in capital, production processes and distribution, each party will be empowered.

Empowerment of the community's economy in the slum areas of the Metro Tanjung Bunga area will have a positive impact on increasing the productivity of economic enterprises, namely: (i) making the community more independent; (ii) helping productive businesses make a large and modern economy; (iii) structural changes in the economy; (iv) the establishment of a good partnership; and (v) encourage the emergence of new entrepreneurs.

## 3. Materials and Methods

### 3.1. Research Design

This research uses a sequential explanatory approach, which is a combination of quantitative and qualitative research methods applied in sequence. Qualitative research is used to investigate, discover, describe, and explain the quality or features of social influences that cannot be explained and described through a quantitative approach [47]. The use of qualitative research in this study is more towards meaning and theory as the basis for formulating a research focus based on the facts that develop in the field. Thus, the researcher acts as the main instrument for data tracking needs in the field. Thus, the researcher acts as the main instrument for data tracking needs in the field. Furthermore, the qualitative approach in this study is used to build a hypothesis that will be proven using a quantitative approach. In accordance with the focus and research objectives to be achieved, the first stage in this study is to use a qualitative approach.

The quantitative research method in this study uses numerical data and emphasizes the measurement of objective results for the needs of statistical analysis. The focus of quantitative methods is to collect data sets and generalize to explain specific phenomena experienced by the population [48,49]. The quantitative approach in this study is used to examine the relationship between the variables under study, in this case the subject under study, the data collected, and the source of the data needed, as well as the data collection tools used in accordance with what was previously planned. The aim is to test the hypotheses that have been formulated in qualitative research. The measurement scale used is an ordinal scale. The main instrument used was a questionnaire. Thus, the quantitative approach in this study is used for descriptive, associative, and correlational purposes.

The incorporation of qualitative-quantitative approaches in this study is carried out with the following considerations: (1) Triangulation logic, in this case the qualitative research results are rechecked on quantitative studies and vice versa, the aim is to strengthen the validity of the findings; (2) Quantitative and qualitative research are combined to provide a general picture; (3) Quantitative research is used on the structural characteristics of social life and qualitative research prioritizes the quality of the subject as a starting point for conducting studies; (4) The quantitative approach is used in analyzing the relationship between changes, while the qualitative approach is used to help align the factors underlying the relationships that are built; (5) The quantitative approach is used to reveal the structural features of large-scale social life, while the qualitative approach is more towards small-scale behavioral, so that when researchers try to reveal the two levels, quantitative and qualitative guidelines are used together; and (6) In order to obtain data from different realities, it is necessary to combine two approaches (quantitative and qualitative). Furthermore, in the implementation of this study using a concurrent triangulation strategy, namely the collection of quantitative and qualitative data carried out simultaneously in one research stage. The combination of qualitative and quantitative research in this study is in Figure 2 below.

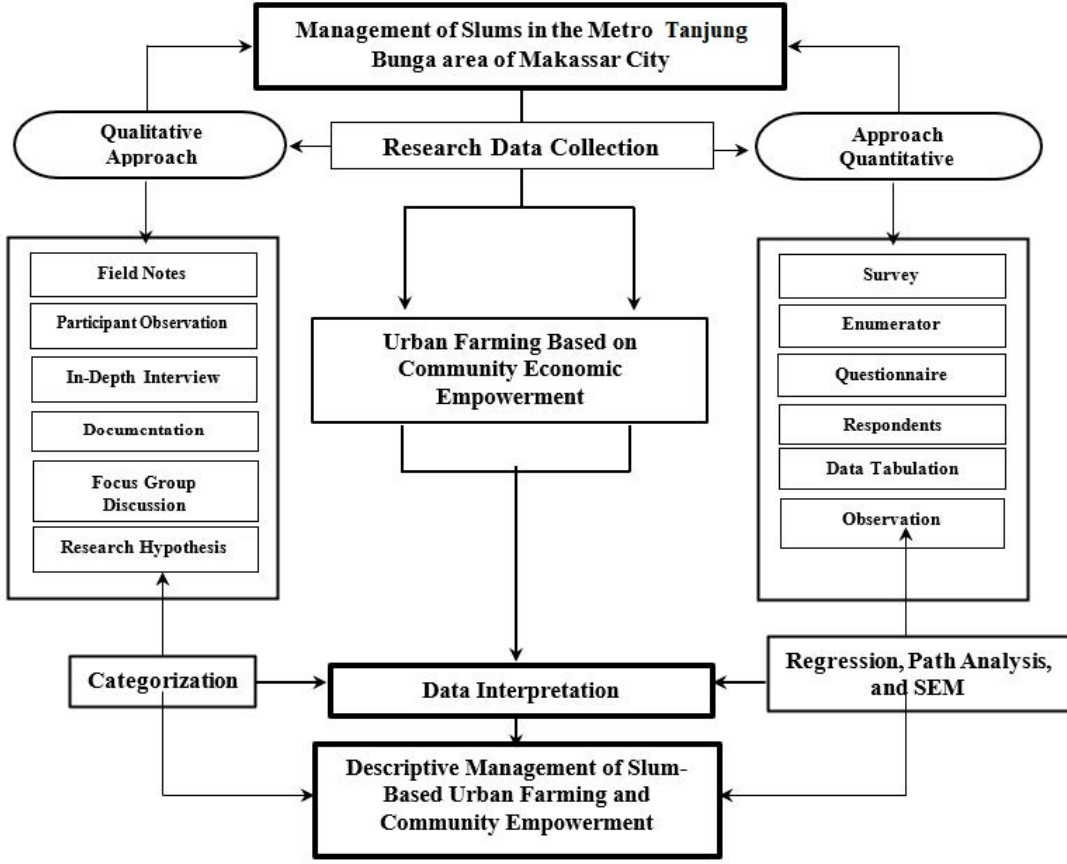

**Figure 2.** Flowchart of the methodology.

### 3.2. Study Area

The urbanization, modernization, and economic growth of Makassar City are determinants of the factors that cause slums to develop (see Figure 1). The selection of the study locations in the Metro Tanjung Bunga area of Makassar City was based on the following considerations: (1) The reality that develops in the field is plural; (2) the inability of the community to adapt and respond to the stimuli of environmental change causes poverty and slum development; (3) changes in community work orientation toward urban industrialization have an impact on marginalization; and (4) the use of space that develops is very complex in terms of the formation of social structure. Furthermore, slums in which the local community live develop as a result, of the rapid change in spatial function, the inability to access economic resources, and changes in environmental stimuli. Urban informal settlements, usually defined by certain criteria such as self-built houses, sub-standard services, and residents' low incomes, are often seen as problematic, due to their associations with poverty, irregularity, and marginalization [35]. The dominant business centers that flourished in the Metro Tanjung Bunga area had an impact on the conversion of productive agricultural land, of land use, and of building functions. Such conversions resulted in the spatial allocation of 48.6 hectares of land for commercial functions, 384.5 hectares for large-scale housing, 55.6 hectares for recreational facilities, 1.57 hectares for education facilities, and 2.1 hectares for health facilities.

The complexity of the dominant commercial use of a space has an impact on the segmentation and separation of residential facilities between migrants and local communities. The greater the social distance (positive or negative) between individuals and the median social position of their environment, the more likely the individual will move from their neighborhood [50]. Furthermore, the current spatial segregation has a tendency that leans toward the typology of change of housing facilities, different income levels, and lifestyle of the community. Conversion of productive agricultural

land and the complexity of spatial use affect the level of spatial separation and produce changes in people's lifestyles [51–56].

This research was conducted in the Metro Tanjung Bunga area of Makassar City. The selection of the research sites was based on considerations: (1) The inability of the community to access the city's economic resources has led to the development of slums; (2) community social capital has the potential to support the application of the concept of urban farming; and (3) urban farming is implemented by utilizing limited land, the use of residential buildings, and spaces open to reducing and controlling environmental degradation. Furthermore, the dominant economic activities carried out by the community are divided into two main categories: (1) Newcomers who predominantly develop formal economic businesses such as shopping centers, tourism, cafes, restaurants, and travel agents; and (2) dominant local communities that develop informal urban activities, namely, food stalls, street vendors, local transportation, and trading businesses in traditional markets.

The characteristics and typologies of the slum areas of the Metro Tanjung Bunga area are divided into three categories: (1) Slum areas located in low-lying and water-edge areas, including Jongaya, Tanjung Merdeka, Balang Baru, Barombong, Maccini Sombala, Mangasa, and Pa'baeng-baeng Villages; (2) slums located in low-lying areas, namely, the village of Manuruki; and (3) slums located in low plains, along the edge of the water, and above the water, namely, the village of Parang Tambung. Furthermore, the population, the size of the slums, and the geographical position of the Metro Tanjung Bunga area by villages are in Tables 1 and 2 and Figure 3 below.

**Table 1.** Number and population density of the Metro Tanjung Bunga area of Makassar City.

| Number | Village | Area (Hectares) | Head of Family | Total Population (Soul) |
|--------|---------|-----------------|----------------|------------------------|
| 1 | Mangasa | 203 | 10,348 | 32,042 |
| 2 | Parang Tambung | 138 | 9749 | 42,396 |
| 3 | Tanjung Mardeka | 337 | 2216 | 11,414 |
| 4 | Barombong | 734 | 3105 | 13,276 |
| 5 | Maccini Sombala | 204 | 5185 | 22,584 |
| 6 | Jongaya | 51 | 3932 | 15,678 |
| 7 | Mannuruki | 154 | 3583 | 12,082 |
| 8 | Pa'baeng-Baeng | 53 | 5059 | 20,731 |
| 9 | Balang Baru | 118 | 4018 | 19,058 |

Source: Makassar City Central Bureau of Statistics [30].

**Table 2.** Geographical location of the slum settlements of the Metro Tanjung Bunga area.

| Number | Village | Slum Area (Hectares) | Geographical Position |
|--------|---------|----------------------|------------------------|
| 1 | Jongaya | 3.2 | 119°25′0.702″ E; 5°10′31.539″ S |
| 2 | Tanjung Merdeka | 17.19 | 119°23′36.434″ E; 5°10′52.923″ S |
| 3 | Balang Baru | 16.31 | 119°24′48.474″ E; 5°11′00.025″ S |
| 4 | Barombong | 30.53 | 119°23′44.642″ E; 5°12′38.569″ S |
| 5 | Maccini Sombala | 24.29 | 119°23′56.486″ E; 5°9′56.429″ S |
| 6 | Mangasa | 22.71 | 119°26′8.623″ E; 5°11′1.453″ S |
| 7 | Mannuruki | 4.16 | 119°25′46.280″ E; 5°10′31.847″ S |
| 8 | Pa'baeng-Baeng | 3.57 | 119°25′26.359″ E; 5°10′16.355″ S |
| 9 | Parang Tambung | 42.84 | 119°25′22.031″ E; 5°11′11.164″ S |

Source: Makassar City Central Bureau of Statistics [30].

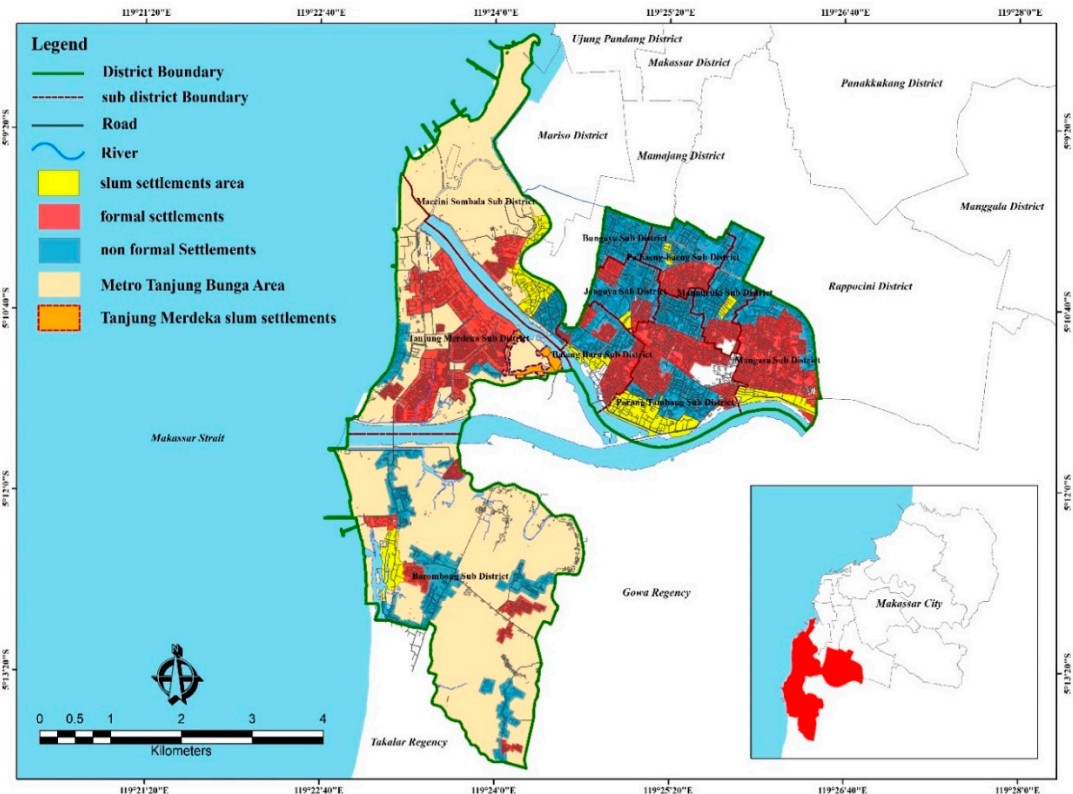

**Figure 3.** The Metro Tanjung Bunga area of Makassar City as the object of research. Source: Authors'elaboration.

*3.3. Data Collection Method*

In qualitative research, the researchers themselves are the main instruments that go into the field in order, to collect data or information through observation, in-depth interviews, and documentation [44,49]. Furthermore, quantitative research according to the two experts, which is the main instrument is a questionnaire which is carried out by collecting data from respondents through surveys. In the implementation of this study, the two instruments were triangulated. Thus, the data collection method in the study is divided into four main categories, namely observation, in-depth interviews, structured interviews, questionnaires, and documentation.

3.3.1. Observation

Observations are carried out to observe the conditions and characteristics of land use in relation to developing spatial functions, and the potential for land that allows it to be used for urban farming needs. Observation is also used to observe the implementation of urban farming for area scale and household scale as part of handling slum settlements in the Tanjung Bunga Metro area. Furthermore, observations in this study use field notes, periodic notes, and checklists. The aim is to provide an overview of the implementation of urban farming based on community economic empowerment.

3.3.2. In-Depth Interviews

The main subjects of this research are the surrounding communities who live in slum areas in nine villages in the Metro Tanjung Bunga area. In-depth interviews were conducted for description and exploration purposes. In-depth interviews are used to build an understanding of the mechanisms and stages of urban farming based on community economic empowerment. In-depth interviews were carried out using tools, namely a tape recorder, pictures, and interview guides equipped with freelance notes, checklists, and a value scale.

### 3.3.3. Questionnaire

The questionnaire in this study is used for two functions, namely (i) descriptive, and (ii) measurement. The purpose of using a questionnaire is to provide an overview of some of the characteristics of the individual who is the object of research. Structured interviews using a questionnaire were conducted by asking questions to the community which was facilitated in the implementation of urban farming based on economic empowerment based on predetermined research variables. Filling in the questionnaire was not submitted to respondents but was guided by the researcher. The selection of respondents in this study was carried out by field workers as well as enumerators. The criteria for the actors who filled out the questionnaire (respondents) were divided into two categories, namely: (1) local communities involved in the implementation of urban farming based on community economic empowerment; (2) local communities with the criteria of staying permanently and not leaving the place for a period of 5 years and having an economic business in the study location.

### 3.3.4. Documentation

A document is a record of events that already apply. This study used various documents relating to the situation and the condition of the Metro Tanjung Bunga area. The documents in question include land ownership status, population data, profiles of local communities, and other documents related to the research objectives. The data collection through documentation was carried out during the study, including land ownership structure documents obtained from the Makassar City Regional Development Planning Agency, data on population numbers obtained from the Makassar City Central Bureau of Statistics, profiles of local communities obtained from the village and district offices, and other documents related to the historical development of the Metro Tanjung Bunga area obtained from the local community.

### 3.3.5. Triangulation

Triangulation is a data collection technique that combines different data from the same source. Triangluation in this study was carried out by combining observation, in-depth interviews, and documentation for data sources simultaneously. The aim is to test the credibility of the data and understand and interpret the implementation of urban farming based on community economic empowerment as part of the efforts to handle slum settlements in the Tanjung Bunga Metro area. The aim is to obtain consistent and thorough data within the framework of achieving research objectives. The data that has been collected through observation, in-depth interviews, questionnaires, and documentation are grouped into two categories of data, namely primary data and secondary data. Sources of data in this study are as follows:

1.  To answer research questions, what influence do business motivation, human resource capacity, community participation, and economic business management have on urban farming-based economic empowerment in the handling of slum settlements in the Metro Tanjung Bunga area of Makassar City? The parameters used as a reference include: (a) business motivation is measured based on the encouragement and willingness of the community, in the sense that strong or weak community motivation plays a role in determining the size of the achievements achieved. Indicators that are used as a reference, namely the driving force, willingness, forming expertise, forming skills, responsibilities and obligations; (b) the capacity of human resources is measured based on the capacity of the community and the role of the institution as a system to carry out its functions or authority to achieve goals effectively and efficiently. Indicators used as a reference, namely competence, education, training, and experience; (c) community participation is measured based on the participation of individuals and community groups to support the implementation and success of urban farming. Indicators used as a reference, namely participation in decision making, participation in the implementation of activities, participation in monitoring, and participation in the utilization of results; (d) economic business management

is measured based on the community's ability to manage an urban farming-based economic business. Indicators that are used as references, namely the type of business, productivity, financial administration management, and business legality; (e) community economic empowerment is measured based on community welfare and the innovations that can be created. Indicators used as a reference, namely the level of income, skills to manage the business, and the stability of the marketing of the products; (f) management of slum settlements is measured based on basic infrastructure services for settlements, community economic conditions, and socio-cultural conditions. The indicators used as a reference are the fulfillment of basic infrastructure services, economic access, and social cohesion.

2. To answer research questions, what are the direct and indirect relationships of urban farming, community capacity, and business capital support for improving the welfare and independence of the community in the Metro Tanjung Bunga area of Makassar City? The parameters used as a reference include: (a) urban farming is measured based on the success of its implementation based on the value of benefits received by the community. Indicators that are measured to assess the implementation of urban farming, namely socialization, activity planning, and activity implementation; (b) community capacity is measured based on adaptability, readiness, and involvement. Indicators that are used as references, namely adaptive response, motivation to attend training, and ability to build a business; (c) business capital support is measured based on the ability to utilize capital, business capital support from the government, and access to finance; (d) community welfare is measured based on the level of income, the ability to meet basic needs, and the ability to pay for the socio-economic needs of the family, namely the ability to build adequate housing facilities, the cost of children's education, and health services. (e) community independence is measured based on awareness and desire to change, ability to increase capacity to gain access, ability to face obstacles, and ability to build cooperation and solidarity. The indicators used as references are community-based development, economic sustainability, community participation, development of social capital, and elimination of socio-economic inequality.

3. To answer the research question, what is the sustainability model for handling slums in the Metro Tanjung Bunga area of Makassar City? The parameters used as a reference, namely (i) infrastructure aspects are measured based on the feasibility of community residential buildings and green open space preparation, (ii) economic activities are measured based on business development and business capital assistance, and (iii) social problem solving is measured based on community participation in settlement of social conflicts and handling slum settlements through optimizing the use of social capital that has been built. Environmental pollution control is carried out through support for changes in attitudes and behavior of the community. The success performance of handling slum settlements is measured based on two things, namely (i) increasing the economic productivity of the community, and (ii) improving the quality of the environment. Furthermore, the sustainability of slum settlement is measured based on three things, namely (i) economic sustainability, (ii) social sustainability, and (iii) community independence.

### 3.4. Research Informants and Respondents

The number of informants in this study was set at 18 people, of which 10 were respondents and the other eight were from outside of the pool of respondents. Those eight non-respondent informants were selected based on information from the village office; the information obtained from the village head was then determined as the base informant. Furthermore, from the base informant's information, the number of informants was determined to total eight people. These informants were then selected using the snowball method, and the final informants were community leaders who still resided in the Metro Tanjung Bunga area of Makassar City.

The researcher then assigned one of the community leaders as the key informant. Next, additional community leaders that could be interviewed were determined, until the information being provided

was essentially the same. In addition, informants (referred to as the perpetrators in the phenomenon under investigation) were also selected from a pool of several respondents that had already been interviewed. The aim was to explore some of the questions to be answered in the questionnaire but that required a more detailed explanation.

The population in this study is the subject under study, while the sample is part of the population studied. The size of the sample in this study depends on the degree of accuracy or error the researcher wants; the error rate used as reference was 5% (0.05). The samples were determined by purposive sampling; In this case the researcher determines the sample based on certain characteristics in accordance with the research objectives so that it can answer the research problem. Purposive sampling is based on the considerations and characteristics of a specific population that have been known before [57–60]. The sample in this study was the head of the family and was determined based on certain characteristics. The characteristics referred to by the researchers to determine the sample are residents who live in slum settlements with families, have lived in the Metro Tanjung Bunga area for at least five years, and understand the mechanisms for handling slum settlements based on urban farming and community economic empowerment. Furthermore, these characteristics are used by researchers to determine respondents. The sampling method according to Cochran [61], using the following formulations:

$$n = \frac{N}{Ndx^2 + 1} \tag{1}$$

where *n* refers to the sample size, *N* refers to the population size, and d refers to the error rate (0.5) or 5% of the 95% confidence level. The number of samples was determined for each village, as shown in Table 3 below.

**Table 3.** Number of respondents by village in the Metro Tanjung Bunga area.

| Number | Village | Population (Head of Family) | Sample (Head of Family) |
| --- | --- | --- | --- |
| 1 | Mangasa | 10,348 | 80 |
| 2 | Parang Tambung | 9749 | 70 |
| 3 | Tanjung Mardeka | 2216 | 20 |
| 4 | Barombong | 3105 | 30 |
| 5 | Maccini Sombala | 5185 | 45 |
| 6 | Jongaya | 3932 | 35 |
| 7 | Mannuruki | 3583 | 35 |
| 8 | Pa'baeng-Baeng | 5059 | 45 |
| 9 | Bungaya | 4018 | 40 |
| Total Samples | | | 400 |

### 3.5. Data Analysis Method

Data analysis was performed by combining qualitative and quantitative analysis in sequence. That is, the steps used for the qualitative research were simultaneously used for the quantitative research. In terms of the interpretation or analysis, the data were then reduced, that is, the qualitative and quantitative data were categorized for statistical calculation. The data is then interpreted using triangulation based on the resources obtained in the field. That is, the data obtained from the results of the questionnaire were explored more deeply through two methods, namely qualitative and quantitative research so as, to strengthen the validity of the results. Furthermore, data reduction was carried out by grouping and categorizing the data in accordance with the research objectives.

Qualitative data analysis in this study used descriptive qualitative methods. The qualitative analysis process carried out included: (1) Analysis of data before going to the field, in this case the researcher used secondary data for the purpose of determining the focus of the study; (2) Data analysis in the field, in this case the researcher uses an interactive model with informants, namely (i) data reduction, in this case choosing the main things and focusing on important things, (ii) presenting the data. Data presentation is done in the form of a short narrative description, (iii) verification.

This analysis was carried out to answer the research objectives, namely urban farming based on community economic empowerment as a solution for handling slum settlements in the Metro Tanjung Bunga area of Makassar City. Furthermore, the quantitative data analysis in this study uses statistical analysis. Regression analysis was used to test the effect of $X_1$ (business motivation), $X_2$ (human resource capacity), $X_3$ (community participation), and $X_4$ (business management) on Y (economic empowerment). Regression methods and correlation analysis were used to answer the first research question. The regression method uses the following formulations:

$$Y = a + b_1X_1 + b_2X_2 + b_nX_n. \tag{2}$$

Y is the dependent variable, X is the independent variable, a is a constant, and $b_{1\times1}$, $b_{2\times2}$, $b_{3\times3}$ are regression coefficients. For the correlation coefficient test, we used the Pearson correlation coefficient (r) considering (i) the research data were interval scale data and (ii) the correlation between two or more variables was linear, meaning that the distribution of the data obtained showed a direct relationship. r was calculated, using the formulation, as follows:

$$rxy = \frac{\sum \varkappa \times y}{\sqrt{(\sum \varkappa^2)(\sum y^2)}} \tag{3}$$

$r\varkappa y$ is the correlation coefficient between variables X and Y, $\varkappa$ is the deviation from the mean for the value of variable X, y is the deviation from the mean for the values for variable Y, $\sum \varkappa \cdot y$ is the product of X and Y values, $\varkappa^2$ is the square of the values $\varkappa$, $y^2$ is the square of the y value. Path analysis is used to answer the second problem formulation. Path analysis was applied based on the following variables: (1) $X_1$ exogenous independent variable (i.e., urban farming); (2) $X_2$ exogenous independent variable (i.e., community capacity); (3) $X_3$ exogenous independent variable (i.e., business capital); (4) Y endogenous dependent variable (i.e., community welfare); and (5) Z endogenous dependent variable (i.e., community independence). Path analysis using structural formulations, as follows:

$$Y = PYX_1 + PYX_2 + PYX_3 + e_1. \tag{4}$$

Path analysis was used with the following considerations: (1) The research metric data use interval scales; (3) there are endogenous dependent and exogenous independent variables for multiple regression models and intermediate variables for mediation models, as well as combined mediation and multiple regression models and complex models; (3) there is a sample size of 400 respondents; (4) the pattern of relationships between variables is only unidirectional; and (5) there is a causal relationship based on theory, that is, there is a relationship or correlation between urban farming, strengthening community capacity, venture capital, and community welfare that influences community independence. The application of the multiple regression and path analyses is shown in in Figure 4.

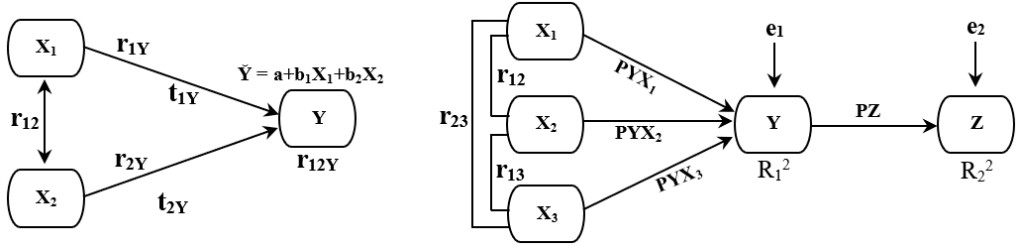

**Figure 4.** Model of multiple regression analysis and path analysis.

Figure 4 shows the multiple linear regression analysis with explanation Y is the dependent variable, $X_1$, $X_2$ is the independent variable, $r_{12}$ is the relationship between $X_1$ and $X_2$, $r_{1Y}$, $r_{2Y}$ is the correlation value between the variables X with Y, $t_{1Y}$ and $t_{2Y}$ are the significance values. In the path

analysis model, several things can be explained, including: (1) The magnitude of the relationship is expressed by the correlation coefficient $(r_{12})$, $(r_{13})$, and $(r_{23})$. $(r_{12})$ shows the correlation or relationship between $X_1$ and $X_2$, $(r_{13})$ shows the correlation or relationship between $X_1$ and $X_3$, and $(r_{23})$ shows the correlation or relationship between $X_2$ and $X_3$; (2) The variables $X_1$, $X_2$, and $X_3$ act as independent variables affecting the dependent variable (Y); (3) The independent variable $X_1$ and the dependent variable Y are connected by the regression coefficient $(p_1)$; (4) The independent variable $X_2$ and the dependent variable Y are connected by the regression coefficient $(p_2)$; (5) The independent variable $X_3$ and the dependent variable Y are connected by the regression coefficient $(p_3)$; (6) the magnitude of the direct effect of $X_1$ on Y is the square of the regression coefficient $(p_{12})$, the direct effect of $X_2$ on Y is the square of the regression coefficient $(p_{22})$, the direct effect of $X_3$ on Y is the squared of the regression coefficient $(p_{32})$; (7) The magnitude of the total influence is the coefficient of determination with the $R^2$ symbol, which is the value of the total effect of the influence of the independent variables under study on the dependent variable: (i) $R^2$ is the total effect, namely the direct effect + the indirect effect, (ii) $(P_{12} + P_{22} + P_{32})$ is the direct effect $X_1$, $X_2$, and $X_3$ on Y, (iii) $(P_1, r_{12}, P_2)$ is the effect indirect variable $X_1$ through $X_2$ on Y, (iv) $(P_2, r_{12}, P_1)$ is the indirect effect of variables $X_2$ through $X_1$ on Y, (v) $(P_1, r_{13}, P_3)$ is the indirect effect of $X_1$ through $X_3$ on Y, (vi) $(P_3, r_{13}, P_1)$ is the indirect effect of variables $X_3$ through $X_1$ on Y, (vii) $(P_2, r_{23}, P_3)$ is the indirect effect of variables $X_2$ through $X_3$ on Y, (viii) $(P_3, r_{23}, P_2)$ is the indirect effect of variables $X_3$ through $X_2$ on Y; (8) Meanwhile, epsilon $(\varepsilon)$ states the amount of residual effect (residue), namely the magnitude of the influence of the independent variables that are not examined.

SEM is used to answer the formulation of the third problem. Structural equation modeling (SEM) was used to build and test the statistical models in the form of causal relationships, which was based on three analyses, namely, factor analysis, path analysis, and regression analysis. Furthermore, consideration was given to interaction modeling, nonlinearity, correlated independent variables, measurement errors, interference errors that correlate with some latent independent variables during the SEM, which were measured using certain indicators. Thus, SEM was used in this study for three purposes: (1) Confirmation of the unidimensionality of various indicators for constructs/concepts/factors; (2) testing the suitability of a model based on empirical data; and (3) testing the suitability of the model and analyzing the causal relationships of the interdependent effects built into the model [62].

The application of SEM in this study was based on the following variables: (1) $X_1$ exogenous independent variables (i.e., infrastructure); (2) $X_2$ exogenous (i.e., economic) independent variables; (3) $X_3$ exogenous (i.e., social) independent variables; and (4) $X_4$ exogenous independent variable (i.e., environment); against (5) $Y_1$ endogenous dependent variable (i.e., performance of slum management); and (6) $Y_2$ endogenous dependent variable (i.e., sustainability of slum management). Next, the indicators of latent and endogenous variable construction included various factors, as follows. (a) Latent variable construction: (1) Infrastructure, measured by the indicators of residential buildings and green open space; (2) economy, measured by the indicators of business development and venture capital assistance; (3) social, measured by indicators of community participation and social capital; and (4) environment, measured by the indicators of community behavior and environmental pollution. (b) Endogenous variable construction: (1) The performance of the slum management program, measured by the indicators of increasing economic productivity and environmentally conscious culture; and (2) the sustainability of the slum settlement management program, measured by the indicators of economic sustainability and community independence. The formulation used and the SEM structural model are shown in Figure 5. SEM analysis method uses the following formulations:

$$\eta_1 = Y_{11}\xi_1 + Y_{12} + \xi_1$$
$$\eta_2 = \beta_{21}\eta_1 + \xi_2$$
$$\eta_2 = \beta_{31} + Y_{32} + \xi_2 + \xi_3$$
$$\eta = B_\eta + \Gamma\xi\xi + \xi \tag{5}$$

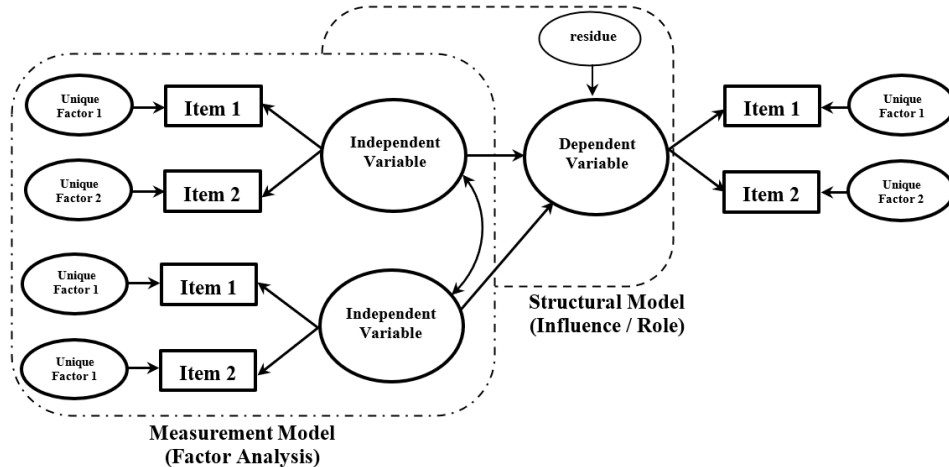

**Figure 5.** Application of structural equation modeling (SEM).

The explanation for $\beta_{21}$, $\beta_{31}$, $\beta n$ is the regression coefficient, $\xi_1$, $\xi_2$ $\xi_3$ is the exogenous variable and $\eta_1$ is the endogenous variable, $X_1$, $X_2$, and $X_3$ are the observed variables, which function as indicators of latent variables $\xi_1$ and $Y_{11}$, $Y_{12}$ function as indicators of latent variables $\eta_1$ and $\eta_2$ residual error (residual term), which is $\zeta_1$ which is related to the prediction of the latent variable value $\eta_1$, the measurement error (error term) for $Y_{11}$, $Y_{12}$, and $Y_{32}$ namely $\varepsilon_1$ and $\varepsilon_2$ and the measurements for $X_1$, $X_2$ and $X_3$, namely $\delta_1$, $\delta_2$, and $\delta_3$. Furthermore, the arrow in the direction of $\xi_1$ to $\eta_1$ indicates that the exogenous latent variable $\xi_1$ affects the endogenous latent variable $\eta_1$. The arrows in the direction of $\xi_1$ to $X_1$, $X_2$ and $X_3$ and from $\eta_1$ to $Y_{11}$ and $Y_{12}$ are represented by $\lambda$. The regression path, which shows the influence of each latent variable on each of the indicators. The two-way arrow between $X_1$ and $X_2$ is the covariance/correlation between the indicators $X_1$ and $X_2$.

The use of SEM in this study was aimed at building a sustainability model for handling the slums in the Metro Tanjung Bunga area. The results of the SEM analysis illustrated two things: (1) A combination of measurement models (factor analysis) and structural models; and (2) the role of the items in measuring the constructs and the role of the constructs against other constructs. Assumptions were built into the SEM, that is: (1) that each indicator has a value that is normally distributed against each of the other indicators; (2) that each variable depends latently on the model that is normally distributed for each value of each of the other latent variables; (3) that there is a linear relationship between the indicator variables and the latent variables, as well as between the latent variables themselves; and (4) that typical indirect measurements, in which case all of the variables in the model, are latent variables.

## 4. Results

The accelerated development of Makassar City, which is predominantly based on economic growth, has an impact on the spatial dynamics and differences in people's interests in accessing economic resources and in decreasing the quality of the environment. Current urban planning practices need to acknowledge these differences to limit that impact on the biosphere while promoting human well-being [63]. Economic growth with regard to the development of commercial space functions is positively associated with an increase in population and an increase in built-up areas, leading to changes in natural land cover and urban flooding, as well as to a decline in the quality of the local community's residential environment. The evolution of the concept of poverty shows an increase in awareness of poverty in certain environments [64]. Furthermore, built-up land is an anthropogenic feature that causes a lack of infiltration of water into the soil, as well as increases the flow above the soil surface. Built-up land is a key indicator in assessing the urban environment [65]. Urban impervious surfaces (UIS) influence the structure and function of urban systems and are widely considered a key indicator of urban environmental conditions [66].

The dynamics of the development of Makassar City in the direction of the Metro Tanjung Bunga area have a direct impact on changes in spatial attributes, as well as a very significant increase in population size. This increase in population size causes social changes and spatial dynamics, leading to ecosystem complexity. To manage multiple ecosystem services (ES) effectively, it is essential to understand how the dynamics of ES maintain healthy ecosystems to avoid potential negative impacts on human well-being in the context of sustainable development [67,68]. Furthermore, urban development is the result of the interaction between anthropogenic and environmental dimensions [69].

Ecosystem complexity due to population pressure continues to increase, along with the increase in large-scale settlement development, services, and commercial activity centers that have a direct impact on reducing land cover and land limitations to prepare public green open spaces. The development of the population in the Metro Tanjung Bunga area of Makassar City is shown in Figure 6.

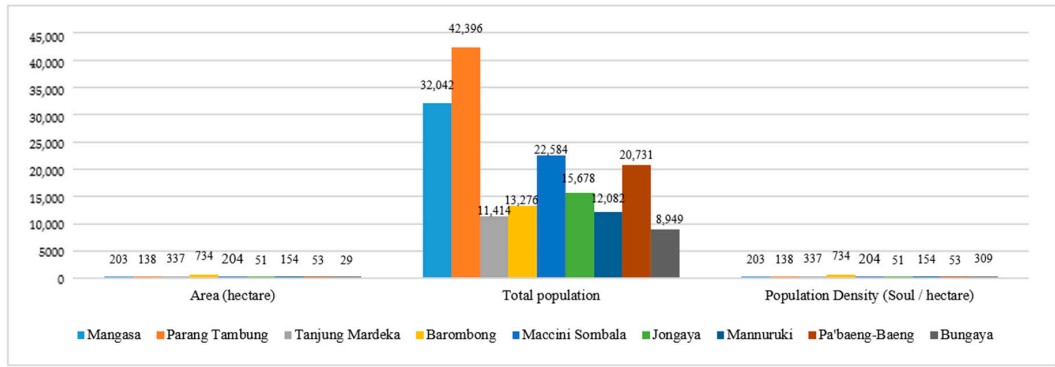

**Figure 6.** Number and population density of the Metro Tanjung Bunga area of Makassar City from 2014 to 2019 [30].

Figure 6 shows the number and population density in the Metro Tanjung Bunga area of Makassar City. Interpretations of this data are as follows: (1) The highest population is located in the village Parang Tambung and the lowest population in the village of Bungaya; (2) the increase in population size is positively associated with changes in social and economic relations of the local community; and (3) changes in spatial use due to an increase in population size have an impact on the powerlessness of local communities in accessing economic resources. The results of the field research illustrate that the powerlessness of the local community is caused by changes in the orientation of agrarian work toward urban industry and differences in interests between migrants and local communities. Job crafting refers to physical and cognitive changes in task or relational work boundaries, enacted by individuals to recreate their work experience in a more motivating and rewarding way, and to realize self-actualization, growth, and meaning at work [70]. Furthermore, work stress is a complex phenomenon that is dynamic and develops over time [71].

The factors that increase the population size in Metro Tanjung Bunga area are as follows: (1) The development of large-scale settlements; (2) the allocation of spatial functions for commercial activities; and (3) the development of tourist facilities. These three urban activities have an impact on the poverty and marginalization of local communities, slums, and environmental quality degradation. Informal settlements and urban informality are serious problems in general in third world countries and in city densities, including areas where environmental impacts occur [72,73].

The process of forming social spaces in the Metro Tanjung Bunga area has a tendency toward changes in spatial characteristics and social systems of the community. The field results show that changes in the dynamics of local communities are influenced by three main factors, that is: (1) Predominantly commercial economic activities have an impact on the creation of a new economic order that requires certain types of expertise and skills in accessing jobs, leading to an index of social differentiation in the livelihoods and education of local communities. (2) The differentiation of urban functions due to land use change and conversion of productive agricultural land results in an increase

in the range of available services, and the effect is very significant on the differences in socio-economic interests. The field results show that the pattern of activities resulted in a dualistic economic system, namely, traditional activities on the one side and modern life on the other. (3) The increasingly complex social order has an impact on population mobility, population composition, and separation of community groups based on ethnicity and economic capacity. These three factors are very important for understanding why some systems and communities either respond weakly or else fail to cope with challenges (exhibiting low levels of resilience); thus, it is imperative to understand how available resources are mobilized and used [74].

Changes in the use of the Metro Tanjung Bunga area in relation to socio-economic dynamics have an impact on the changes in social formation from the previous condition, namely, traditional agriculture to urban industry. Changes in social formation exhibit the characteristics of social life, from traditional primitive traits to modern lifestyles. The results of the field confirmation show that the social change of the local community is positively associated with changes in the scale of people's lifestyle. The three categories of spatial expressions related to lifestyle changes, namely, the social level, urbanization, and segregation. Each of these expressions reflects a dominant tendency in social organizations. In subsequent developments, the impact on the poverty and marginality of local communities is due to the inability to respond to changes in environmental stimuli. Consequently, poverty and inequality are among the first objectives of sustainable development and are then consolidated into its social aspect, without which balanced development is not possible [75].

## 4.1. Poverty and Community Marginalization

Public poverty is synonymous with the inability to access economic resources. The rapid and revolutionary development of the Metro Tanjung Bunga area and the weak control of spatial use have a direct impact on community segmentation. This condition is marked by the separation of local community housing facilities, which initially constituted a unified community group and dominantly sought economic activity in the agriculture and fisheries sectors. The field findings show that after the Metro Tanjung Bunga area was built, leading to large-scale commercial activities and settlements, it caused a decrease in the productivity of agricultural land, and the social relationships of local communities experienced significant changes, resulting in economic relationships. Furthermore, the developing economic relationships have an impact on the differences in interests and in meeting the needs of the local community. The hierarchy of meeting the needs of local communities is assessed based on five main levels: (1) Physiological needs, (2) security needs, (3) social relationships, (4) self-esteem, and (5) self-actualization. These five things have a direct influence on the changes in the use of space in the Metro Tanjung Bunga area. The hierarchy of the fulfillment of the needs of the local communities is shown in Figure 7.

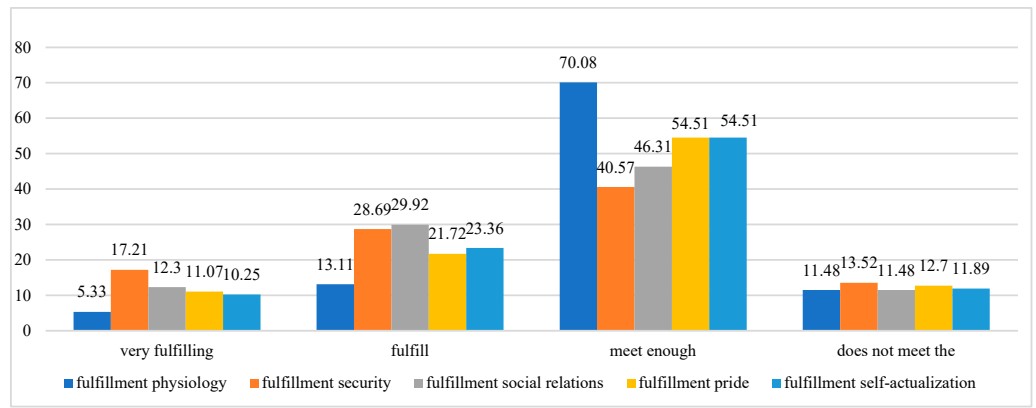

**Figure 7.** Hierarchy of the fulfilment of the needs of the local community in the Metro Tanjung Bunga area of Makassar City. Source: Primary data.

Figure 7 shows the existence of a local community in the Metro Tanjung Bunga area. The proposed interpretation is related to the hierarchy of the fulfillment of local communities, due to changes in spatial use that are very fast and revolutionary. First, the relationship of changes in spatial use with the fulfillment of the physiological needs of the local community showed that 18.44% were able to meet their basic needs, meaning 81.56% of the local community were unable to meet their basic needs. This illustrates that changes in spatial use have an impact on poverty and the marginalization of local communities. Thus, government policy support is needed to protect the existence of local communities by increasing business productivity and should be implemented through economic empowerment mechanisms. Political empowerment has a direct positive effect on economic and managerial empowerment, but the effect is not significant with regard, to social empowerment [76].

Second, the relationship between changes in spatial use and the fulfillment of the security needs of local communities shows that 45.90% mentioned meeting these needs, and as much as 54.10% said that their security needs were not meet. The field results indicate that changes in spatial use leading to a sense of security only occurred for local communities working in formal urban activities. Third, the rapid and revolutionary changes in spatial use have an impact on social relationships between migrants and local communities. As many as 42.22% said that the social relationship between infiltrative migrants and the local community is symbiotic mualistic. Furthermore, as many as 57.78% mentioned that their social relationship needs were not met. The results of the field confirmation show that positive social relations only occur between the local community and the presence of infiltrative migrant populations and are in the same housing environment. Migration, in other words, and the ability and capacity of communities to deal with it responsibly, is a function of community sustainability [77].

Fourth, the rapid and revolutionary changes in spatial use are positively associated with differences in the fulfillment of the self-esteem needs of local communities. As many as 33.79% mentioned that they felt proud to be citizens of the Metro Tanjung Bunga area, meaning as many as 66.21% mentioned not fulfilling their self-esteem needs. That is, pride as citizens is only felt by some local community groups. Fifth, changes in the use of space have a positive contribution to the fulfillment of the self-actualization needs of the local community. As many as 33.61% mentioned fulfilling, this needs while as many as 66.39% mentioned not fulfilling it. This means that changes in spatial use do not have a direct positive effect on the development of economic ventures developed by local communities.

The above-reviewed five aspects illustrate that changes in spatial use have a positive contribution to the existence of local communities. This positive contribution only occurs in the social relationships between the local community and the infiltrative migrant population. Furthermore, the four other aspects tend to weaken or do not have a direct relationship with meeting the hierarchy of needs of the local communities that still survive in the Metro Tanjung Bunga area. Thus, landscape stewardship is increasingly understood within the framing of complex socio-ecological systems [78].

Poverty and marginalization of the community in the Metro Tanjung Bunga area are marked by a change in orientation of agricultural activities toward the work system of urban industrial society. The acceleration of the development of Makassar City is characterized by the disparity of urban intercultural services that are positively associated with the quality of space reproduction control [79]. This includes the economic restructuring and the release of redundant workers previously hidden inside the workplace, as well as the increasing migrant population who are excluded from the formal urban institutions [80]. The slums that are formed are characterized by unsuitable building conditions and infrastructure facilities and settlements that do not meet service standards, thus becoming a threat to the environmental health of the local community. The rapid growth of densely populated, predominantly low-income residential areas in the cities constitutes a serious threat to health [81].

The increased poverty rate, which tends to keep rising, is due to changes in spatial use and the accumulation of various commercial space functions that are predominantly developed in line with the influx of migrants to the Metro Tanjung Bunga area. Furthermore, the local community space previously utilized for productive agricultural activities has undergone spatial and physical changes, leading to the

formation of a very complex social space structure. Such rapid urbanization will ultimately result in multi-scalar separation, which involves the separation of time scales and space scales [82]. Changes in the dominant use of space that result in commercial functions lead to segmentation, differences in community interests, and socio-economic segregation. Segmentation and segregation then develop into different lifestyles, housing facilities, and settlement infrastructure services between newcomers and the local communities. The typology of luxury homes inhabited by capital-intensive groups is facilitated through government and private policy collaboration [83]. The growth of slums has major consequences for both humans and the environment, resulting in the residents of slum areas experiencing a variety of substandard, dense, and unhealthy housing conditions [84].

## 4.2. The Characteristics and Typology of Slum Settlements

Slums are basically multi-perspective and are very difficult to define. It is very difficult to define slums as a result, of covering all aspects that are interrelated with one another [85,86]. Having no formal definition in place for identifying slums adds yet another level of subjectivity to defining slums [21]. Herein, the characteristics and typology of slums in the Metro Tanjung Bunga area were determined by referring to the Law of the Republic of Indonesia No. 1 of 2011. The parameters used as a reference to determine the typology and characteristics of said slums were: (1) buildings, (2) environmental roads, (3) environmental drainage, (4) environmental sanitation, (5) fire protection, (6) clean water services, and (7) suitability of the location for city spatial plans.

Figure 8 shows the typology and characteristics of local community slums in the Metro Tanjung Bunga area. Such slum settlements are divided into two main categories: (1) The typology of settlements is generally categorized as being located in lowland, waterfront, and above-water areas; and (2) the level of slums consists of two categories, namely heavy slums and moderate slums. The results found in the field illustrate that land use values undergo very significant changes. Furthermore, the tendency of the changes in spatial use shows that the existing land use value is no longer assessed based on the productivity of agricultural land, but rather is judged based on the economic function of space. The closer the distance to the location of economic activity, the higher the land value; conversely, the further the distance from the location of economic activity, the lower the land value. These results illustrate that the weak community access to spatial functions and the high value of land have an impact on environmental degradation, poverty, and slum development. Thus, government policy support is needed to solve slum-related issues, which are important for maintaining sustainable development and adequate urban planning [87].

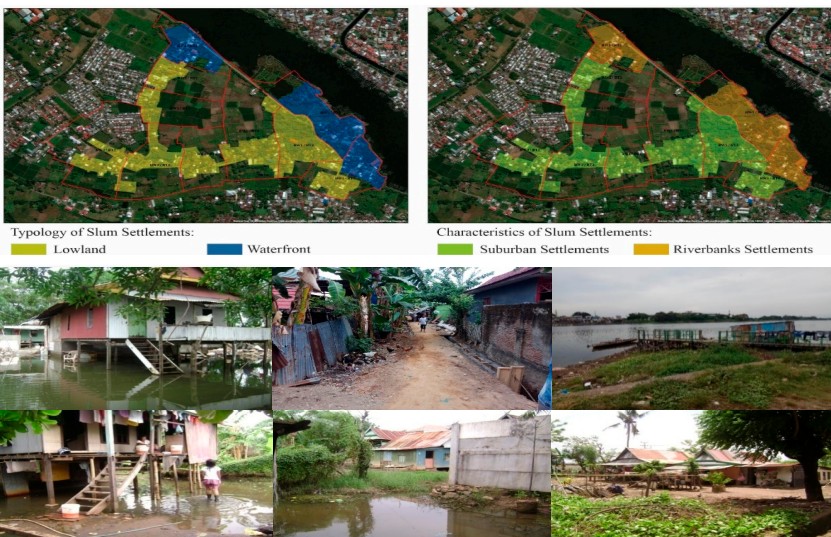

**Figure 8.** Example of the typology and slum levels of the Metro Tanjung Bunga area of Makassar City. Source: Authors' elaboration.

The typology of slum settlements is determined based on the conditions and characteristics of the location, in this case the slum settlements in the Metro Tanjung Bunga area are predominantly located in riverbank and coastal areas. Meanwhile, the determination of the characteristics of slum settlements is measured using three main indicators, namely: (1) location criteria based on a single unit of housing and settlement entities experiencing environmental degradation; (2) The condition of a residential building with a high density, irregular and not meeting technical requirements; (3) Facilities and infrastructure conditions that do not meet the standard of settlement services, namely environmental roads, environmental drainage, clean water services, waste management, waste water management and fire protection.

The existence of slums is a consequence of building densification in the Metro Tanjung Bunga area. Furthermore, building densification and weak control of spatial use are determinants of the development of slums. Furthermore, disparity in socio-economic status impacts on the powerlessness of local communities to access economic resources. The quality of slums is severely impacted in terms of their housing facilities and inadequate infrastructure conditions. The field results show that the location of local community settlements that are in direct contact with elite settlements are not supported by adequate road infrastructure or drainage networks, making them very vulnerable to the threat of flooding. Another supporting fact is that developers in the implementation of development have a tendency, to close access to the location of local community settlements.

The decline in environmental quality due to building density, population density, and low awareness of the community has a high impact on the potential pollution of river flows and the residential environment. Likewise, the transmission of infectious diseases is very easy due to the intensity of social contact. In addition, the residence of the local community is also very vulnerable to the threat of fire, due to the density of the building and the high level of building density, making it easier for the fire to spread from one house to another. In the process of urbanization, policymakers should pay more attention to health costs and regional differentiated management, as well as to more widely promoting the construction of healthy cities [88].

The allocation of the newly developed housing space by the developer is quite intensive, causing existing local communities to break through and utilize land along the riverbank for residential facilities. The potential role of income and the difficulties in slums contribute to the burden of mental illness, and they are a result of the government's failure to provide adequate and affordable shelter to the urban poor [89,90]. Rising environmental costs, growing social exclusion, and intra-urban are clear threats [91]. The status change of the ownership from local communities to migrant populations facilitated by developers and government policies has an impact on marginalization, poverty, and the development of slums on illegal lands. Labor inflows threaten employment opportunities and reduce income for local labor [92]; the direct impact of this process is the subsequent high criminal rates, increased unemployment, and decreased environmental quality along riverbanks, thus leading to river water pollution. The main factors that influence the criminal behavior of potential offenders include unemployment, poverty, poor governance, and weaknesses in law enforcement or crime control institutions [93].

The pattern of the development of settlements in the vicinity of luxury housing complexes whose uniform shape, area, architectural style, and building quality in terms of price directly filters an influx of migrants to the Metro Tanjung Bunga area. The field results show that differences in housing facilities that develop between migrants and local communities have an impact on physical, economic, and social segregation. Three indicators form the basis for justifying these conditions: (1) Occupancy ownership is based on income, (2) occupancy ownership is based on socio-economic class, and (3) selection and ownership of housing locations are based on ethnic groups (migrants and local communities). These results illustrate that the very complex variation of social space causes physical separation of space between migrants and local communities. The political and socio-economic backgrounds of different periods have had a great influence on the urban socio-spatial structure [94]. Furthermore,

the very intensive transfer of land use causes changes in land ownership status from local community ownership to developer ownership, as shown in Figure 9.

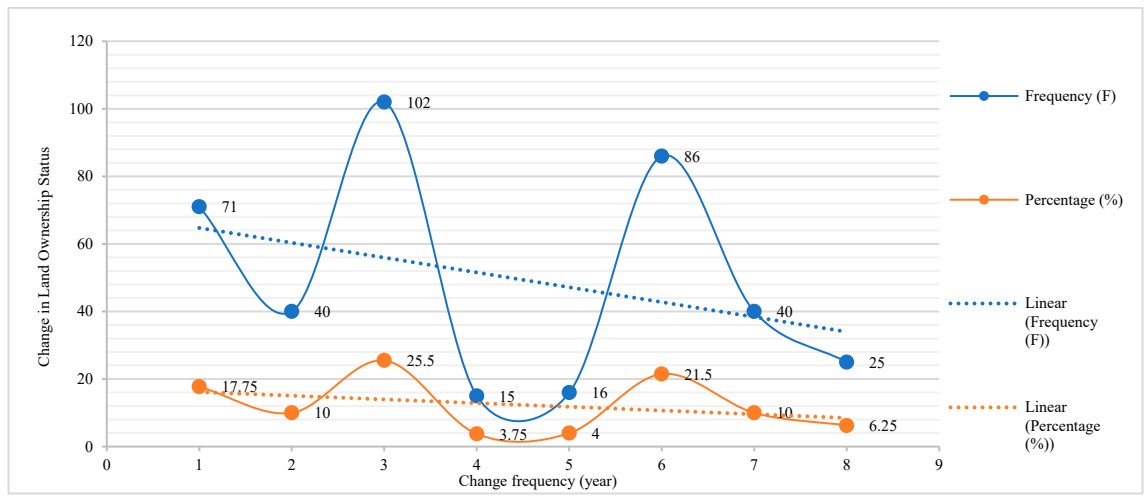

Source: Primary data.

**Figure 9.** Change of land ownership status of the local communities in the Metro Tanjung Bunga area of Makassar City.

Figure 9 shows the change in land ownership status that was initially dominated by local communities, then experienced a shift toward developer ownership. The process of the transfer of land ownership between local communities to developers that took place in the Metro Tanjung Bunga area is divided into several stages: (1) The first stage or initial period took place from 1994 to 1995 with a total of 71 (17.5%) transactions between the local community and developers; (2) the highest level of transfer took place between the period of 2001–2003 with 102 (25.50%) transactions; (3) the period of 2004–2006 experienced the lowest level of transfer, with 15 (3.75%) land sale and purchase transactions; and (4) the last stage was 25 (6.25%) land sales transactions between 2004 and 2006 and 6.56% between 2007 and 2009. The results of the in-depth interviews highlight that the factors that caused these high land purchase transactions were due to the value of land in the Metro Tanjung Bunga area in the 2001–2003 period, which was quite low due to the lack of support in accessibility. Accessibility is a pivotal element in this context, because it may induce increases in land value, whereby some or all of these increments in land value that result from the increase in accessibility can be captured to recover the capital costs of a transport investment [95].

The fact that the field results illustrate that the relatively low land value in the Metro Tanjung Bunga area has prompted developers to carry out massive land acquisition. Furthermore, the intensity of the land acquisition has become the driving force for the acceleration of development and changes in spatial use. This process is positively associated with reducing the area and productivity of agricultural land, and the change of social relationships into economic relationships causes a decrease in kinship relationships in local community groups. The next impact is fragmentation in the life of community groups, which leads to the separation of residential facilities between local communities which initially resided in a unified residential environment. Land use decisions may direct urban expansion toward certain guidelines favoring collusion between landowners [96].

Status changes in land tenure due to the development of large-scale settlements and commercial space functions are positively associated with the inability of local communities to adapt to environmental situations. The field results indicate that some of the local communities affected by such development then manage to break through into the empty space, particularly along riverbanks, and use them to build residential facilities with modest material conditions, thus developing illegal slums. Thus, slums are the by-product of social and economic impacts due to rapid urbanization [97].

Slums that develop along riverbanks have an impact on land cover (i.e., a reduction), water pollution, and the potential threat of flooding. The cause of such environmental degradation is the decreasing amount of land cover along the river course zone [98]. Thus, it can be concluded that the construction of large-scale settlements and the function of commercial space cause poverty, marginalization, and the development of slums. Slums pose a significant challenge for urban planning and policy as they provide shelter to a third of urban residents [99]. Furthermore, the composition of the environment and access to employment implies that the relocation program must be carefully designed if it is to improve community welfare [100]. Thus, the continuation of several ideas, namely, capacity building, slum dialogue, community participation, and infrastructure development are very important in the process of settling slums [101].

### 4.3. The Socio-Economic Characteristics of Local Communities

The community characteristics in the slums of the Metro Tanjung Bunga area are divided into two categories based on the socio-economic conditions, that is: (1) The subsystem economy includes people who are only able to meet their basic needs in a limited way, with an average income level of $65–100 per month; and (2) the commercial economy includes people who are able to meet their basic needs and send their children to school, but that have limited ability to obtain health services, with an average income level of $150–250 per month. Income level is strongly associated with the cultural system of society and the ability to adapt to environmental changes. In other words, culture tends to change—although, in certain parts, its traditional characteristics are static—but culture has a dynamic side so as, to adapt in line with environmental changes [102].

Figure 10 shows the level of income, the work that can be achieved, and the work done by the local community to seek work in the Metro Tanjung Bunga area of Makassar City. Three outcomes were identified (Figure 10A): (1) 18% of local community income is high; (2) 17.2% is moderate; and (3) 64.75% is low. The field results illustrate that the difference in income levels is very much related to the adaptability of local communities to changes in environmental stimuli. On the other hand, local communities who are unable to adapt to the changing situation of spatial use are in a marginal position, categorized as poor people, and their housing facilities are categorized as slums.

Furthermore, the work achieved by the local community (Figure 10B) illustrates that as many as 7.25% of the local community choose work as shopkeepers. A further 10.25% work at the center of economic activity, 35.25% work as domestic servants in elite housing, 26.50% as construction workers on housing being built, 14.25% as parking attendants, and 6.50% as a security across elite housing, shopping centers, and tourist activities. The complexity of the work of the local community, which was initially in the dominant agricultural sector but later impacted urban industry, has led to changes in social strata, social status, and diverse social classes, which were initially relatively homogeneous and later developed in a heterogeneous direction according to specifications, expertise, and achievements.

The results of the in-depth interviews with informants reinforce this fact by highlighting that, in the Metro Tanjung Bunga area, it is very difficult to rely on agricultural activities or to obtain maximum income to meet the necessities of life, given the development of agricultural lands into housing and urban activity areas. Thus, changes in spatial use and land use change that are very intensive become a driving force of change in the work orientation of local communities. This means that the reduction of agricultural land runs parallel to the rationalization of the actions of individual local communities from the peasant profession to other professions from time to time.

To secure job opportunities in the Metro Tanjung Bunga area requires adequate expertise, skills, and educational background. The field results show that not all local communities are able to fulfill these requirements. The efforts made by several local community groups (Figure 10C) show that as many as 29.75% mentioned that being able to work requires adequate formal education; 24.25% mentioned that it requires skill and expertise in certain fields; 31.25% mentioned being able to work in the function of economic activities requires training and courses that too must be supported by an adequate formal educational background. Furthermore, 14.75% mentioned others, requirements—in

this case, that connections and recommendations from certain parties are needed. The results of the in-depth interviews support this statement by confirming that to work in economic activities, there is a minimum level of formal education that must be met, as well as the need for recommendations from certain parties. Thus, it can be concluded that changes in the work orientation of local communities have an impact on social mobility, both vertically and horizontally.

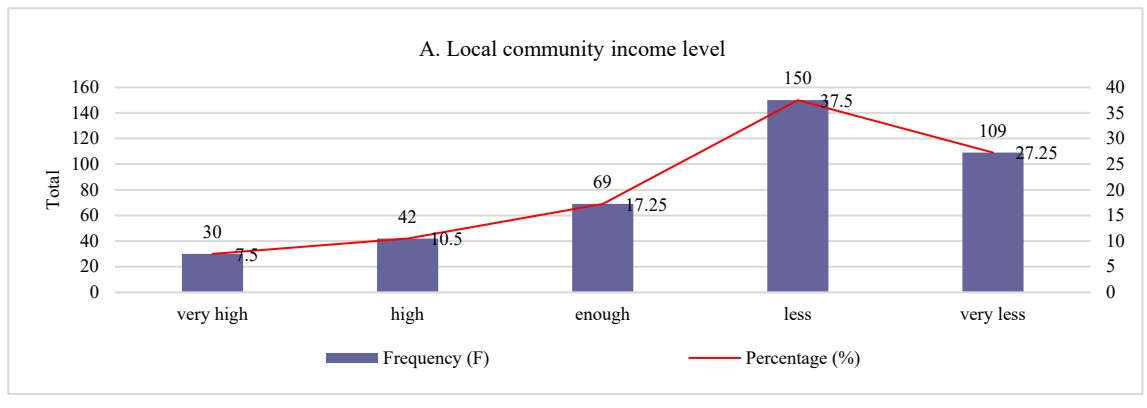

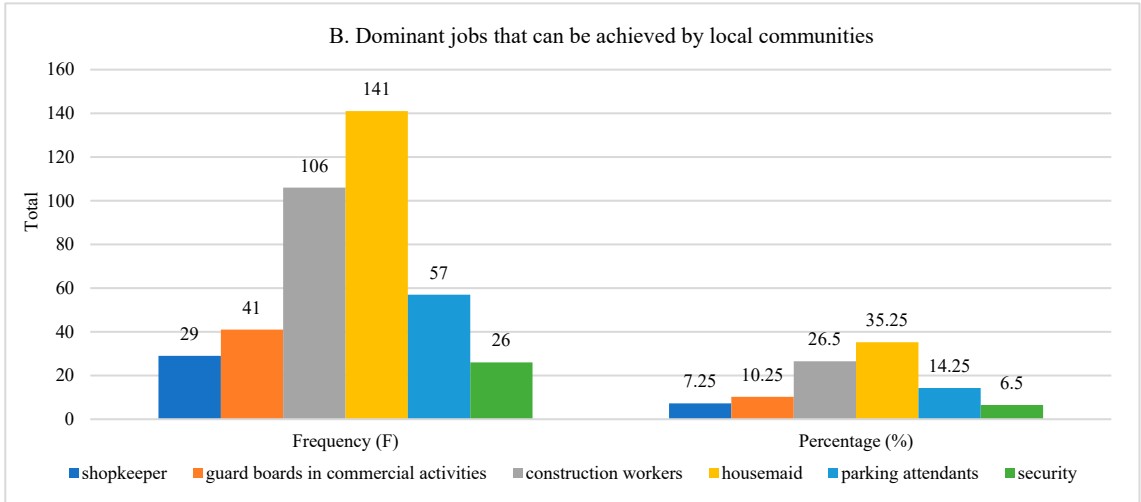

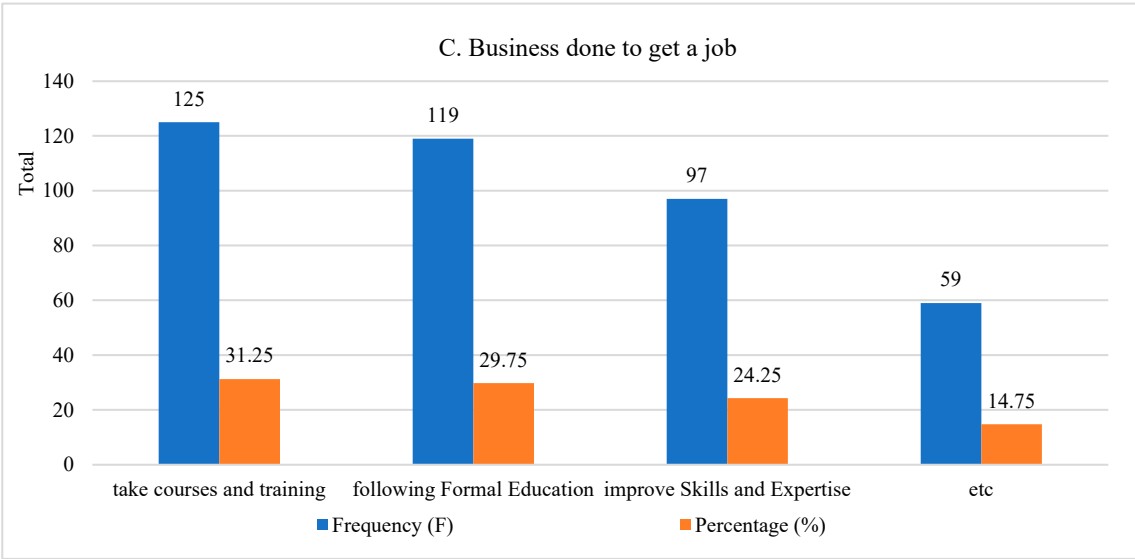

**Figure 10.** Example income levels, jobs that can be achieved, and efforts of local communities. (**A**) Local community income level, (**B**) Dominant jobs that can be achieved by local communities, (**C**) Business done to get a job.

Differences in housing facilities, economic endeavors, and work orientation between migrants and local communities show different abilities and different levels of welfare of the community, thus causing the formal and informal economies to develop toward a dualistic economic system. Low socio-economic groups significantly differ in their direction of movement, group separation, and differences by means of developing settlements in the suburbs [4]. Furthermore, the relationship between the migrant population and the established local community is divided into two categories: (1) Employer–worker relationships. This pattern shows that social relations are built by utilizing local communities as labor that is built by migrants. (2) Social relationships in the neighborhood where the local community lives. This pattern is built for the benefit of maintaining kinship and family ties, leading to mutually beneficial relationships based on mutual interests in achieving family welfare and survival. As a social potential developed in local communities, it is used to preserve the existence of said local communities under ever-changing environmental conditions [103]. The pattern of relationships that are created is very dependent on the intensity of interactions and social distance. In this case, resilience is considered a necessary condition of urban society as a socio-ecological subsystem of urban ecosystems [104].

The strength of interactions highly depends on the ability of individual migrants and local communities to adapt to the evolving environmental situation and spatial functions. For local communities located in slums, it is marked by the inability to build an economic business, thus requiring support and facilitation in the form of economic empowerment. The main factors affecting economic resilience are the availability of social communities and formal economic institutions, institutional capacity, availability of production centers, and social capacity [105,106]. Furthermore, to develop the economic business of the community requires capital support from the government. Interventions to help reduce urban poverty include: (1) Programs aimed at improving living conditions, mainly through improving slums, but also through housing projects and the provision of public sites and services, thus enabling access to housing credit and financing, land certification, infrastructure improvements, and utility subsidies; and (2) programs that aim to increase the income of the poor, such as job training and micro-business development [107]. The public perceptions related to the need for venture capital to increase economic business productivity are shown in Figure 11.

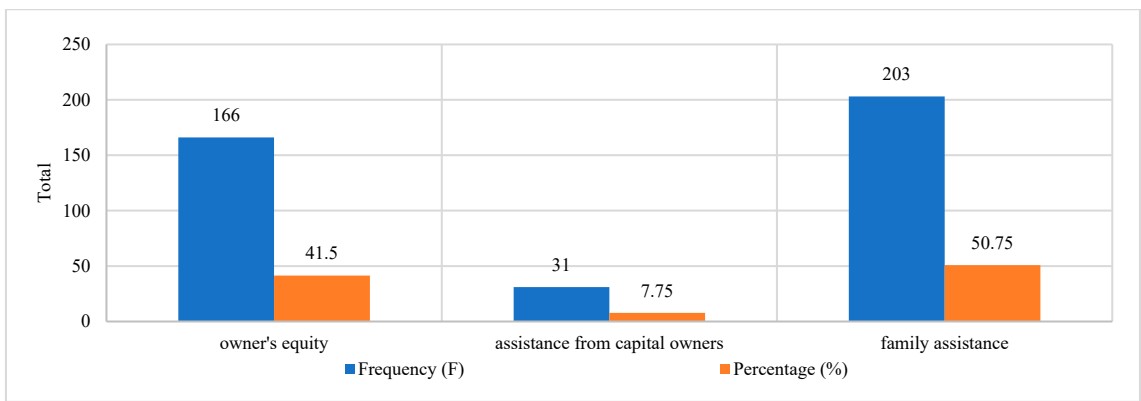

**Figure 11.** Capital used to develop local community economic enterprises.

Figure 11 shows the ability of local communities to develop business activities and to capitalize on capital. The following results are noted: (1) 41.50% of the local community stated that use their own capital from the sale of land to develop economic ventures; (2) 50.75% stated that they family assistance as their business capital; and (3) 7.75% mentioned receiving business assistance from capital owners. The field results support these facts by confirming that the business activities developed by the dominant local community are based upon their own capital and family assistance. The reason is that using one's own capital does not incur interest, in addition to venture capital sourced from the family, which is counted as debt without interest and is not limited by a repayment period, but is instead based on the profits obtained using a profit sharing system.

Business capital support is very important for the poor and marginal communities located in the slums in the Metro Tanjung Bunga area. Business capital support is achieved through facilitation, assistance, financial management, and community capacity-building, which are carried out through business development, access to capital, and marketing of economic business results. The implementation of this policy was developed to foster public confidence in government program policies and was implemented by building networks with collaborative patterns between government, the private sector, and the community. This policy encourages public confidence in the government by applying the principles of social justice and equitable development, and by reducing disparities to lead to stability in development [108,109]. Furthermore, the recognition and legitimacy of local communities determines the overall initial success in community collaboration [110].

The handling of slums is not only focused on infrastructure completion, but also requires a strategic program whereby the implementation is comprehensive and collaborative, including physical, economic, social, and cultural factors. Effective collaboration between central government, local governments, and communities is critical for ensuring smooth program execution and accountability [111]. Furthermore, the local governments are expected to form economic coalitions with developers and political coalitions with the central government [112]. Economic empowerment is an effort made to encourage the improvement of community welfare in the slums of the Metro Tanjung Bunga area. Community economic empowerment has a significant effect on improving community welfare [113]. Furthermore, urban farming based on economic empowerment is oriented toward the creation of new jobs and is considered a solution to overcome poverty and unemployment to increase the welfare and productivity of economic businesses. Urban agriculture also includes livestock, aquaculture, agroforestry, and horticulture. In the broadest sense, urban agriculture describes the entire system of food production that occurs in cities [114].

*4.4. The Implementation of Community-Based Urban Farming Concepts*

Agricultural land conversion due to the increased intensity in the development of commercial space functions and large-scale settlements is part of the urban development system and its associated impacts. The concept of urban farming innovation basically refers to the concept of sustainable development [115], meaning that the implemented process mechanism refers to the triple bottom line concept, which is the balance between aspects of profit, people, and the planet. The concept of urban farming that is implemented in slums in the Metro Tanjung Bunga area includes: (1) Agricultural systems that utilize yards and vacant land through plant cultivation technology with aquaponic methods as an alternative to grow plants and to raise fish in one container; and (2) hydroponic patterns that are used by considering multiple factors, namely, more efficient and environmental land use, planting throughout the year, higher production rate, use of fertilizers and lack of pesticides, more efficient water use, and shorter planting periods.

Figure 12 shows an example of the implementation of urban farming as part of the handling of slums in the Metro Tanjung Bunga area of Makassar City. Four urban farming concepts are applied to support the improvement of the local community's economic endeavors. First is the use of vacant land (Figure 12A,B) that is in direct contact with the Jenneberang watershed and local community slums. The established concept patterns include: (1) The construction of a greenhouse as a laboratory that, at the same time, serves as a medium for plant and fish breeding (2) the use of a system that combines aquaculture and hydroponics in a symbiotic environment, the purpose of which is to manage the excretion of fish and the increase in water toxicity; and (3) cost efficiency, in the sense that this combination is carried out to obtain optimal benefits and to make it easier for the community to obtain plant and fish seedlings and to facilitate the water needs of such activities.

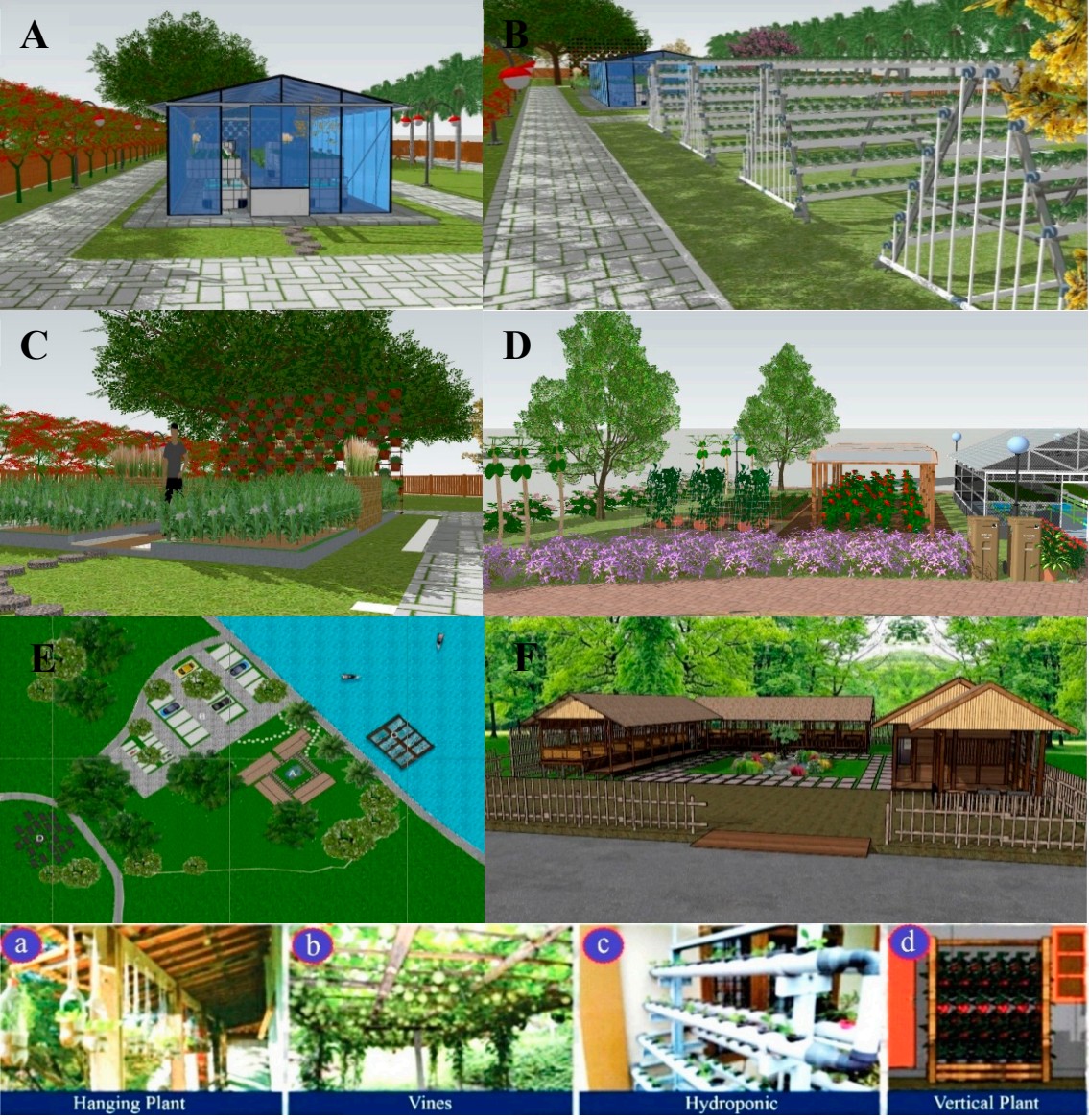

**Figure 12.** (**A**–**F**): Example of urban farming based on community economic empowerment in the slum settlements of the Metro Tanjung Bunga area of Makassar City. (**a**) Hanging Plant; (**b**) Vines; (**c**) Hydroponic; (**d**) Vertical Plant.

Second is the land use for economic business interests (Figure 12C,D); for example: (1) to develop green open space oriented toward three principles, namely, esthetic function, socio-cultural function, and economic function; (2) to develop of different plant types, such as ornamental plants and protective vegetation suitable for watersheds; and (3) combined with biopores with a diameter of 10 cm and a depth of 100 cm. The aim is to prevent flooding in the residential environment, increase groundwater reserves, and treat household waste to not pollute the watersheds.

Third is the integration of the urban farming system into the tourism activities in the Metro Tanjung Bunga area (Figure 12E,F). The goal is not only to market the results of community businesses at harvest, but also to accommodate the traditional restaurant businesses developed by local communities. The basis for considering the application of this concept is that, in addition to the community developing an urban farming business, a restaurant business is also developed, where sufficient raw materials are available, easily obtained, and produced by the local community.

Fourth is the development of urban farming on a household scale by utilizing limited availability of yards (Figure 12a–d). Here, four patterns are applied: (1) An aeroponic system, with the aim to provide maximum and fast results. This condition is very possible if the nutrient solution that is given is sprayed in the form of mist and directly at the roots, so that the roots of the plants can more easily absorb the nutrient solutions, which contain lots of oxygen; (2) a drip system, by using a timer that serves to control the pump. The goal is that when the pump is turned on, it distributes nutrients into each plant. Furthermore, to stand tall, plants are supported by using other plant media such as peat, roasted husk, and sand, which can be easily sourced in the local area; (3) a nutrient film technique (NFT), for application by residents who have a large amount land in order to continuously to drain the nutrients dissolved in water without using a timer for the pump, as the nutrients flows into the gully through the roots of plants and then back again into the water reservoir; and (4) a wick system, implemented using a very simple method intended for people who are beginners in using a hydroponic system. This system is passive because there are no moving parts; nutrients flow into the growth media from the container using a type of wick.

Urban farming implementation is divided into the following three management categories: (1) A group system, applied to vacant land, ranging from nurseries through to the planting, harvesting, and marketing carried out by the community themselves in groups, accommodated by the Community Self-Help Agency (BKM) and community self-help groups; (2) individually, applied at the household scale by utilizing limited land; and (3) the marketing system of production, carried out in four ways, namely, in collaboration with groups, by supplying restaurant needs and catering needs, selling directly to the market, and using technology (IT) as a means of buying and selling transactions. Furthermore, the mechanism adopted in the implementation of urban farming in the Metro Tanjung Bunga area is through the process of mentoring, coaching, and facilitation in the form of economic and social empowerment.

The forms of empowerment that are carried out begin with the socialization stage, moving on to the introduction of concepts, the selection of fruit and vegetable horticulture commodities, the application process, entrepreneurial patterns, financial management, business capital support, and the marketing system of production. These stages of the empowerment process are expected to be able to encourage an increase in the productivity of community economic ventures in the sense that increasing the welfare and independence of the community will have an impact on reducing unemployment, poverty, and the sustainability of economic enterprises through the support of the utilization of social capital that has been built up in people's lives. Human capital, including social capital is the central determinant of resource productivity and sustainability [116]. Thus, the implementation of urban farming becomes part of the solution to the handling of slums in the Metro Tanjung Bunga area of Makassar City. The concept of urban farming implemented in the handling of slums based on economic empowerment in the Metro Tanjung Bunga area aims to encourage the improvement of economic productive endeavors in order, to increase the community income and independence in terms of sustainability. The practice of urban agriculture has gained importance due to the rising rate of urban poverty and population size in developing regions [33]. Urban policies need to incorporate food security considerations and should focus more on building cities that are more resilient to crises [117].

The results of the regression analysis of the tested variables (see Table 4) illustrate that: (1) The relationship between business motivation and economic empowerment is 0.980 and the coefficient of determination of the influence of business motivation is 0.960; (2) the relationship between human resource capacity and economic empowerment is 0.980 and the coefficient of determination of the influence of business motivation is 0.960; (3) the relationship between society participation and economic empowerment is 0.973 and the coefficient of determination of the influence of business motivation is 0.945; and (4) the relationship between business management and economic empowerment is 0.977 and the coefficient of determination of the influence of business motivation is 0.954. Thus, it can be concluded that business motivation, human resource capacity, community participation, and business

management together have a significant effect on community economic empowerment in the handling of slums.

**Table 4.** Summary of the results of the associative hypothesis testing.

| Correlated Variables | *t* Count | Sig. | Information | R$^2$ |
| --- | --- | --- | --- | --- |
| Business motivation leading to economic empowerment | 3.932 | 0.000 | Significant | 0.980 |
| Human resource capacity leading to economic empowerment | 3.848 | 0.000 | Significant | 0.980 |
| Community participation to economic empowerment | −3.962 | 0.000 | Significant | 0.973 |
| Business management to economic empowerment | 2.680 | 0.000 | Significant | 0.977 |
| Business motivation, human resource capacity, community participation, and business management leading to economic empowerment | 10.646 | 0.000 | Significant | 0.968 |

The implementation of urban farming in the slums of the Metro Tanjung Bunga area is divided into three main categories (see Figure 12), which are: (1) Limited land using a greenhouse system. Greenhouses are complex thermodynamic systems, where indoor temperature and humidity must be monitored closely to facilitate plant growth and production [118]. The concept of hydroponics applied in aquaculture by using water without using soil media and more emphasis on meeting the nutritional needs of plants. The process is also combined with freshwater fish farming and its handling is carried out in groups. (2) Utilization of vacant land for the needs of green open space combined with vegetable horticulture plants and equipped with sports facilities and playgrounds. (3) At the household scale, planting is carried out individually and develop vegetable horticulture with a hydroponic system. The types of vegetables include mustard greens, cauliflower, tomatoes, chilies, and vines, which blend with housing facilities and use a drip irrigation system. Furthermore, a greenhouse was developed for the purpose of controlling the microclimate control system by utilizing a sprayer network at the top. The parameters such as air temperature, humidity, $CO_2$ concentration, light levels, air movement, pH, and osmoticum can be manipulated to some extent to regulate crops on a time schedule and to obtain economic production [119].

### 4.5. Economic Sustainability and Community Independence

The sustainability of the community's economic endeavors in the Metro Tanjung Bunga area of Makassar City is oriented toward increasing the productivity of said economic ventures, thus strengthening the capacity and independence of the community. The increased awareness that community and institutional life have shared qualities can do better to increase community capacity, to manage risk, to accept change, and to seize opportunities [120]. Economic empowerment for local communities is focused on several stages, namely: (1) Understanding the concept of urban farming, carried out through the stages of socialization—in this case, the community understands the benefits of implementing urban farming (i.e., increasing economic added value and increasing income); (2) strengthening community capacity, carried out through training, guidance on access to capital, utilization of funds to support innovation, and creativity to develop economic business products and product marketing systems; and (3) supporting venture capital, carried out through government support to facilitate initial capital that can be utilized by the community to develop a business and to market of the products produced.

Consistency of these three principles will encourage improvement of the welfare and sustainability of community economic efforts in the slums of the Metro Tanjung Bunga area of Makassar City. Sustainable entrepreneurship will be able to generate employment, improve products and processes, and establish new companies, thus changing people's lives [121]. Within this context, entrepreneurship is increasingly being recognized as a significant conduit for bringing about a transformation to sustainable products and processes, with numerous high-profile thinkers advocating entrepreneurship as a panacea for many social and environmental concerns [122]. Furthermore, development should

meet the needs of the present without reducing the ability of future generations to meet their own needs [123].

Community economic empowerment through the implementation of an urban farming program is an integration of systems related to strengthening community capacity, which is facilitated through business capital support in order to improve the welfare and independence of the community located in the slums of the Metro Tanjung Bunga area of Makassar City (see Figure 13). The results of the path analysis show the following: (1) The relationship between urban farming and community capacity strengthening is 0.820 (i.e., 82.0%); (2) the relationship between urban farming and business capital support amounts to 0.602 (i.e., 60.2%); and (3) the relationship between community capacity-building and capital support is 0.740 (i.e., 74.0%). Meanwhile, the direct effect of urban farming on improving people's welfare is 0.276 (i.e., 27.66%); the direct effect of strengthening community capacity on welfare improvement is 0.559 (i.e., 55.95%); and the direct effect of business capital support on improving community welfare is 0.367 (i.e., 36.72%). Thus, it can be concluded that urban farming, community capacity-building, and business capital support all have a positive effect on improving community welfare in the handling of slums.

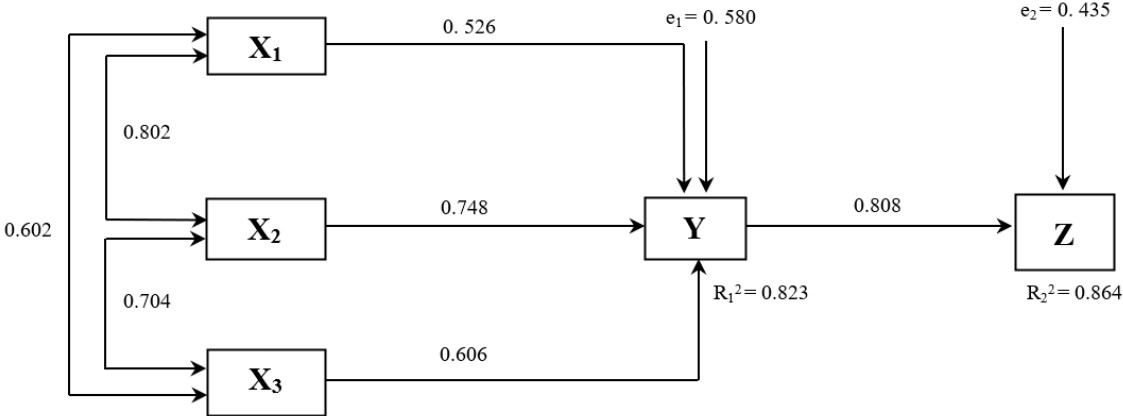

**Figure 13.** The relationship between urban farming, strengthening community capacity, and supporting business capital in relation to improving community welfare and independence.

Additionally, the indirect effect of urban farming through strengthening community capacity to increase community welfare is 0.236 (i.e., 23.68%); the indirect effect of strengthening the capacity of the community through the support of venture capital to increase community welfare is 0.315 (i.e., 31.55%); the indirect effect of urban farming through business capital support to increase community welfare is 0.224 (i.e., 22.44%); the indirect effect of business capital support through urban farming is 0.1918 (i.e., 19.18%); the indirect effect of strengthening community capacity through business capital support is 0.3191 (i.e., 31.91%); and the indirect effect of business capital support through strengthening community capacity is 0.037 (i.e., 3.71%). The total indirect effect is 33.64%, and influence or residue (i.e., the effect of other variables on improving the welfare of the community that is not examined) is 0.6636 (i.e., 66.36%). Furthermore, the direct effect of business capital support on improving community welfare is 0.653 (i.e., 65.3%), and influence or residue amounts to 0.347 (i.e., 34.7%). Thus, it can be concluded that urban farming, community capacity-building, and business capital support contribute positively to improving the welfare and independence of the community. The sustainability in handling community-based slums is outlined in Figure 14.

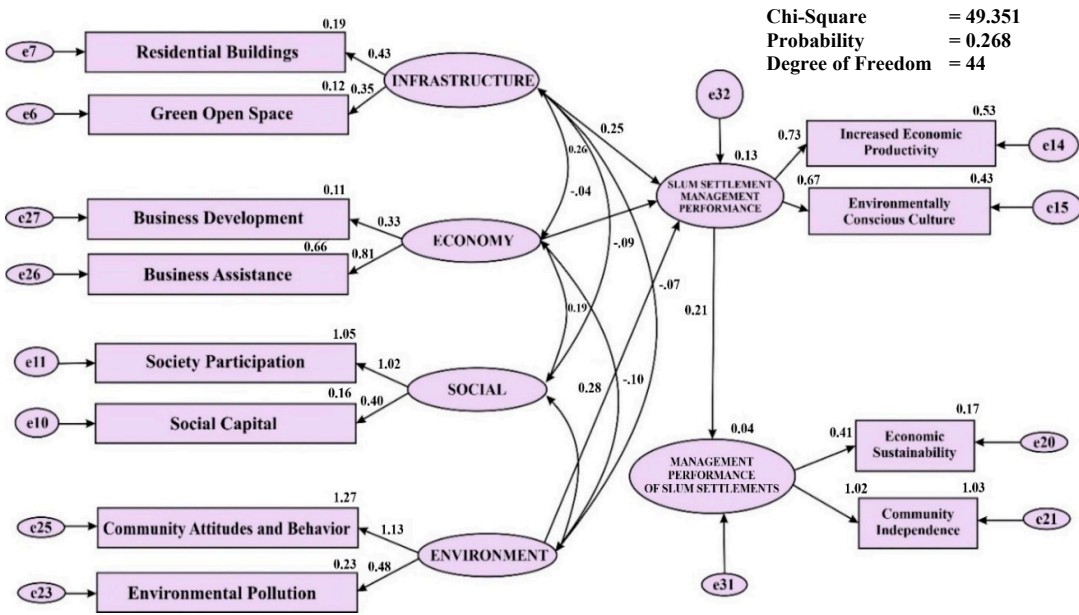

**Figure 14.** Sustainability in the handling of the slum settlements of the Metro Tanjung Bunga area.

Figure 14 shows that the infrastructure (latent variable), economy, and environment significantly determine the performance of slum management programs, but the performance of such management programs does not significantly affect the sustainability of slum management. The structural model (see Figure 14) statistically qualifies as a model to illustrate the program of handling slums in the Metro Tanjung Bunga area, with $p = 0.268 > 0.05$ and df = 44. Furthermore, the effect of the total economic, environmental, social, and infrastructure construction on the endogenous variables of the performance of the slum management programs is −0.141, 0.625, 0.000, and 0.001, respectively. The effect of the total economic, environmental, social, and infrastructure construction on the endogenous variables of the sustainability of slum management programs is −0.011, 0.049, 0.000, and 0.000, respectively. Furthermore, the effect of the total endogenous variables on the performance of slum management programs of the other endogenous variable, namely, the sustainability of slum management programs, is 0.078.

The direct effects of economic, environmental, social, and infrastructure development on endogenous variables of slum management program performance are 0.141, 0.625, 0.000, and 0.001, respectively. This shows that the indirect effect is the same as the total effect, assuming that there is no indirect effect on the model designed. The same is true for the performance of slum management programs towards the sustainability of slum management. Thus, the indirect effect on the structural model that was designed in terms of the contribution of exogenous construction and endogenous construction (i.e., performance of slum management programs and sustainability in handling slums) is zero.

The development of sustainability in the handling of the slums in the Metro Tanjung Bunga area of Makassar City in the future should include infrastructure handling, the development of economic potential, and community participation and the creation of social cohesion. In the context of handling slums, it is very important to maintain the balance of the ecosystem as a manifestation of human embodiments, the natural environment, and the living environment as part of social, economic, and environmental development. The determinants underlying sustainability are the availability of resources, the ability to adapt (i.e., flexibility), homeostasis, response capacity, independence, and empowerment [124]. Furthermore, regional environmental assessment (REA) has been shown to be effective in supporting local sustainability development processes [125].

The handling of slums as part of the development of sustainability will require government policy support, including: (1) Economic equality through increasing the productivity of community economic

efforts; (2) creating social cohesion through efforts to optimize the use of social capital owned by community groups as an integrated urban system; and (3) the management of residential environments through efforts to control pollution and to develop a culture of environmental awareness in people's lives. Diverse land uses are needed to provide a good balance between the provision and non-provision of ecosystem services [126].

The sustainability of handling slums in the Metro Tanjung Bunga area is reliant on several actions, such as: (1) To create a balance of green ecosystem-based development and green open space preparation for the purpose of maintaining the existence of water absorbing areas and for securing the areas that benefit from the river; (2) to realize economic equality by restructuring the production systems of community economic enterprises using corporate social responsibility (CSR) funds to save natural resources and energy; (3) to guarantee social justice in terms of social services and the distribution of wealth; and (4) to create a living environment that is comfortable and safe in order to realize zero emissions. With the development of ecological science, the demand to integrate ecosystem services into ecological management is increasing. Stakeholders are interested in comparing stocks and the ability to supply ecosystem services in different regions [127]. The success of sustainable development does not only depend on the economic sector, but also requires government support, funding collaboration, achieving equity, community participation, justice, and prosperity. Indeed, the provision of ecosystem services from multifunctional landscapes has already contributed to human well-being [128].

## 5. Discussion

The acceleration of the development of the Metro Tanjung Bunga area has basically resulted in an expansion of the Makassar City area toward the periphery through a spatial centrifugal process. The impact of such as expansion causes changes in land use and in the conversion of agricultural land that is very intensive and reflects variations in the intensity of spatial use. There are six factors that drive this process: First, the accessibility factor. The change in accessibility began with the construction of the Metro Tanjung Bunga road corridor toward the Gowa and Takalar Regencies. This accessibility is supported by the development of transportation facilities and infrastructure, which has led to increased inter-regional and inter-city accessibility of the Metro Tanjung Bunga area and has resulted in changes in the use of transportation modes, where boat facilities were initially the only mode of transportation. The dynamic growth of the Metro Tanjung Bunga area due to very intensive land use changes has an impact in terms of reducing agricultural and aquacultural areas as a result, of highly complex urban land use. The complexity of spatial use causes changes in typology and morphology, the socio-economic system of the community, and the cultural patterns of rural agrarian societies in the direction of urban industries. Rapid urban development and expansion bring about significant migration from rural areas to cities for non-farming occupations [129–132].

Second, public service factors drive population mobility and change the urban functions of the Metro Tanjung Bunga area. The results of the analysis show that the pulling factor for the influx of migrants is triggered by the presence of such establishments as educational facilities, health facilities, recreational facilities, large-scale settlements, and shopping centers. This is the driving force for the influx of expansive and infiltrating migrant populations in the Metro Tanjung Bunga area. Third, land characteristics also trigger the high intensity of development, as well as the development of the functions of urban activities. The physical characteristics of land are assessed based on the following three main indicators: (1) A relatively flat topographic condition—in this case, an altitude of 0–3 m above sea level; (2) a small slope (0–5%), in the sense that the micro-relief is not difficult for development, which triggers the development of the functions of new activities; and (3) the availability of sufficient land for the development of large scale settlements and other socio-economic activities. Another influential factor is distance—in this case, the Metro Tanjung Bunga area is very easily accessed from the center of Makassar City. Urban expansion may result in a dense form, in which new urban areas are located near existing urban structures [133].

Fourth, the characteristic factor of land ownership basically refers to the pattern of spatial development in relation to the acceleration and intensity of development. Existing land ownership, if controlled by a group of people or individuals who have a strong economic status, will be different from land ownership owned by those with a weak economic status. The existence of local communities is generally categorized as having a weak economic status and is most easily affected by rising land prices. On the other hand, local community land management efforts are not optimally profitable in relation to the economic productivity of land. The relatively lower economic status of local communities has caused a high rate of land purchase transactions, in addition to factors affecting the level of accessibility and land value. Land and people are the foundation of every nation, while in urban areas, rapid economic and social development are exerting sustained pressure on land demand [134].

Fifth, the existence of regulations that govern spatial planning is one of the factors that influence the intensity of changes in spatial use in the Metro Tanjung Bunga area. In 1997, the function assigned to the Metro Tanjung Bunga area designated it as an agricultural and fishpond area (primary function), and its supporting functions were tourism and settlement. Furthermore, in the 2003–2010 period, the function of the Metro Tanjung Bunga area underwent a very fundamental change in the Makassar City development policy, resulting in new established functions that changed its primary function to become a commercial, service, and tourism center, with settlement and agriculture supporting functions. Thus, inconsistent development policies contribute positively to the conversion of land use functions, particularly productive agricultural land, leading to very complex spatial use.

Sixth, developers' initiatives are a factor in directing spatial physical development and changes in space utilization in the Metro Tanjung Bunga area. The existence of developers has a significant and dominant influence in the process of land use conversion, particularly the conversion of productive agricultural land. The transfer of land ownership status from local communities to developers has an impact on typological and morphological changes. This condition is marked by large-scale housing developments built by developers and the subsequent impact on environmental degradation, poverty, slum growth, and the formation of very complex social territory structures.

### 5.1. Urban Farming-Based Slum Settlement Solutions

The handling of slums based on urban farming and community empowerment in the Metro Tanjung Bunga area is a concept developed for the purpose of encouraging increased economic productivity and community self-reliance in order to reduce poverty and unemployment rates, and to improve the quality of the local community slums. This concept has a strategic value to economic growth and food security so as, to support the future development of Makassar City. Furthermore, the participation of stakeholders and the government is very important in the application of the concept of urban farming, particularly in the process of formulating development policies targeted at food security, community economic security, and the fulfillment of green open space in a sustainable manner. Urban agriculture can contribute to feeding city dwellers, as well as to improving metropolitan environments by providing more green open space [135]. Furthermore, urban–rural relationships are an integral part of development in both urban and rural communities [136].

The implementation urban farming and community empowerment in the handling of slums, in addition to creating economic opportunities for local communities, also plays an important role in fulfilling green open space at the scale of the urban environment, private green open space at the scale of household units, flood control, adding to the esthetic value of the residential environment, controlling environmental pollution, and adapting to global climate. Urban green space has been promoted as an approach to respond to major urban environmental and social challenges, such as reducing the ecological footprint, improving human health and well-being, and adapting to climate change [137]. Furthermore, the implementation of slum management is not only focused on economic empowerment but is also carried out through social empowerment mechanisms.

The handling of local community slums in the Metro Tanjung Bunga area is a unified socio-economic empowerment system; for example: (1) Proactively building community participation; (2) optimizing

community social capital; and (3) creating social cohesion within the community. Furthermore, these three things require the support of several actions, namely: (a) Enabling the community; (b) changing people's behavior; and (c) organizing the community. These three patterns have a direct impact on changes in people's attitudes and behavior in responding to various forms of environmental change, and they also encourage increased activity of the community in the development process. Thus, community empowerment will increase the ability of the community to analyze conditions, potentials, and problems that need to be addressed independently. The use of the empowerment expansion model has proven to be an applicable, relevant, simple, and inexpensive tool for the evaluation of community empowerment [138].

*5.2. Sustainability of Community-Based Slum Settlements*

The orientation of the handling of slums in the Metro Tanjung Bunga area as a unified system includes: (1) Infrastructure aspects, focused on handling the feasibility of community residential buildings and preparing green open spaces; (2) business activities of the community's economy, focused on three important things, namely, business development, venture capital assistance, and the support of the marketing system for the production results; and (3) settlement of social problems, focused on two main things, namely, increasing community participation in resolving social conflicts and optimizing the use of social capital that has been built for the creation of social cohesion, which require collaborative support from various parties. This support includes: (1) Support of government policies in making decisions to side with the poor to solve infrastructure, economic, and social problems; (2) private sector support—in this case, the role of the developer to facilitate and provide venture capital support for the poor located in slums and to directly impact the development of the Metro Tanjung Bunga area; and (3) full community participation in responding to various programs in the framework of promoting sustainable welfare improvement.

The strategic roles of the above-mentioned three elements, namely, the government, the private sector, and the community, will contribute positively to improving the quality of the environment of slums, leading to the integration of the urban system and the economic recovery of the community, thus creating sustainable social justice. Cities face various adversities and challenges, such as the unsustainable use of natural resources, the lack of housing and infrastructure, the prevalence of poverty, rapid urbanization, crime, disasters, and effects of climate change [139]. Thus, sustainability is a concept that needs to be applied to all human activities, especially those within cities, and to the extent that physical aspects, the environment, low-quality buildings, infrastructure, and public facilities can be handled optimally through the role of the government, the private sector, and the community as a whole system [140,141].

Furthermore, controlling environmental pollution will require the support of the community in terms of changes in attitudes and behavior in order, to overcome the problems of the environment in which they live. This means that environmental pollution control will require cooperation from various parties—in this case, the recovery of slum environments, leading to improvements in environmental health, pollution and garbage levels, clean water services, environmental sanitation, flood control, and sustainable river pollution control. The local government is not only an implementer of development agenda, but so are policymakers in their forming of rules of how to connect global goals with the local community [142].

The indicators of the success of the performance of handling slums in the future include the following two main things: (1) Increasing community economic productivity; and (2) improving the quality of environmental settlements through an environmentally conscious culture. Furthermore, the handling of the slums in the Metro Tanjung Bunga area of Makassar City in the future requires two main things, namely, economic sustainability and development that is oriented toward community independence. Thus, system integration is needed in the development of infrastructure, economic empowerment, and social empowerment, in the control of environmental pollution, and in the improvement of environmental quality based on strengthening the institutional capacity of the

government and society. In addition, the existence of developers in the Metro Tanjung Bunga area is not only responsible for environmental conditions but must also provide social benefits for the surrounding community in the form of community empowerment programs.

## 6. Conclusions

The accelerated development of the Metro Tanjung Bunga area of Makassar City toward economic growth is in line with urbanization, large-scale settlement development, and commercial space functions, which all cause an increase in population size, the maximum density, the conversion of productive agricultural land, the level of poverty, and the growth of slums. The existence of slums that a predominantly located on riverbanks and in coastal areas results in a greater risk of flood and environmental pollution. Urban farming based on economic empowerment is oriented toward the creation of new jobs and is considered a solution to overcome poverty and unemployment so as, to improve the community welfare, economic business productivity, and community independence. The results of the regression analysis confirm that business motivation, human resource capacity, community participation, and business management all have a significant effect on community economic empowerment in the handling of slums. Furthermore, the results of the path analysis confirm that urban farming, community capacity-building, and business capital support contribute positively to the improvement of the welfare and independence of the community.

The mechanism adopted in the implementation of urban farming in the Metro Tanjung Bunga area is through the process of mentoring, coaching, and facilitation in the form of economic and social empowerment. The forms of empowerment that are carried out begin with the socialization stage, moving on to the introduction of concepts, the selection of fruit and vegetable horticulture commodities, the application process, entrepreneurial patterns, financial management, business capital support, and the marketing system of production. These stages of the empowerment process are expected to be able to encourage an increase in the productivity of community economic ventures in the sense that increasing the welfare and independence of the community will have an impact on reducing unemployment, poverty, and the sustainability of economic enterprises through the support of the utilization of social capital that has been built up in people's lives.

The implementation urban farming and community empowerment in the handling of slums, in addition to creating economic opportunities for local communities, also plays an important role in fulfilling green open space at the scale of the urban environment, private green open space at the scale of household units, flood control, adding to the esthetic value of the residential environment, controlling environmental pollution, and adapting to global climate. Furthermore, controlling environmental pollution will require the support of the community in terms of changes in attitudes and behavior in order, to overcome the problems of the environment in which they live. This means that environmental pollution control will require cooperation from various parties—in this case, the recovery of slum environments, leading to improvements in environmental health, pollution and garbage levels, clean water services, environmental sanitation, flood control, and sustainable river pollution control.

Sustainable development in the context of handling slums is realized through several strategic actions, such as economic access for poor and low-income people, the creation of social cohesion, the management and control of environmental pollution, and the building of a culture of environmental awareness. The sustainability of the handling of slums is oriented toward the creation of a balance of green ecosystem-based development, preparation of green open spaces, securing riverine benefits, restructuring the production system of economic enterprises, saving natural resources and energy, and ensuring the distribution of wealth and social services, as well as toward community participation in realizing a comfortable and safe living environment so as to achieve zero emissions.

**Author Contributions:** B.S., S.S., and H.H. compiled the research; B.S., H.H.S., and B.B. designed the methodology; A.T.F., and S.S. processed the data; B.S., H.H.S., and H.H. contributed to the materials/methodology/analysis tools; B.S. analyzed the data; A.T.F. contributed to the examination of the data; B.S., S.S., H.H., B.B., A.T.F., and H.H.S. wrote and revised the draft. All authors have read and agreed to the published version of the manuscript.

**Funding:** This research is funded by the Government of the Republic of Indonesia through the Ministry of Research and Technology in the form of development research grant assistance, with grant number: 035/LPPM/UNIBOS/VI/2020.

**Acknowledgments:** We are grateful for the participation of stakeholders in contributing ideas in carrying out this study. Thank you to the Ministry of Research and Technology of the Republic of Indonesia for their support and financial assistance in carrying out this research.

**Conflicts of Interest:** The authors declare no conflict of interest.

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
