# Peer review of "Management of Slum-Based Urban Farming and Economic Empowerment of the Community of Makassar City, South Sulawesi, Indonesia"

_sustainability, doi:10.3390/su12187324_

Round 1

Reviewer 1 Report

The paper is interesting but requires improvement.

The English is poor and would benefit from a native language proof reader or editor. 

General comments

No where in the manuscript do the authors state what definition of empowerment they use. Several different forms of empowerment are used in the text including but not limited to "empowerment" as stand alone term, "economic empowerment", "community empowerment", "community economic empowerment", "urban farming and community empowerment" (with hyphens in different places), "social empowerment", "political empowerment", "economic and managerial empowerment".  Moreover, the authors seem to be using different empowerment terms interchangeably in some places.

It would help if the authors give a clear definition of empowerment and limit the terms to the key relevant ones.

The authors stress the importance of geo-spatial factors. I concur with that observation, but in their analysis there is a notable lack of use of geo-spatial variables.

In the analysis a variable is used for economic empowerment, but the authors fail to identify how that is measured. Moreover, it is unclear how the quantitative and qualitative data are matched and used consistently.

The same holds true for sustainability. Sustainability is not defined, but used without proper explanation in the regression model.

The paper is quite long and contains a lot of detail and information, this means the reader will get lost very quickly. Either the paper needs to be reorganized so that it is clear how the narrative runs, or the paper should be shortened focusing a single key aspect the authors would like to raise. 

More specific comments:

In section 3.4 the variable names are used multiple times for different concepts. Better to use unique variable names.

Many of the variables used in the regression model require proper explanation how they were captured in the data. I already mentioned empowerment and sustainability bt this also holds true for welfare, community independence, community capacity, urban farming)

For each of the variables provide summary statistics.  

Figure 5 contains elements that are not in the methodology section see line 288.

There are numerous typographical errors, some are quite annoying because it makes interpretation of the sentence difficult (e.g. "usea" in line 440)

The description of the SEM model is very cryptic indeed, I have no clue what the elements in the model mean, definitions of for instance ξ, η, ζ and  Γ are not provided.

The relationship between the methodology section and the results as weak. I expect to have a clear description of the methods and tools used in the methods section with clear arguments for the choice of tools and approaches. In the results section I expect to see the results of the approaches highlighted in the methods section. 

What I also miss is a brief description of what approaches are used to answer the three main research questions. In the results it would also help to clearly delimit what results are linked to what research question, how and why.

In Figure 7 and the text around it, fulfillment is presented without any proper explanation in the methods section how this is measured. 

The authors mention the relationship between spatial use and fulfillment but not spatial analytics methods are used.

 Is the typology in section 4.2 ad hoc or based on rigorous clustering statistics, if the latter is the case how was it done. If the former is the case, what is the justification for the typology.

What is in Figure 9, there is no definition of the x-axis and the y-axis is not unambiguous.

The middle part of Figure 10 is not legible. 

I also expect the conclusions to delve more deeply into what the study implied for each of the research questions.

As I mentioned earlier the narrative has to be more clear with methods aligned to the research questions and reflected in the results that lead to conclusions that answering the research questions.

Author Response

Dear Reviewer

We hereby thank you for suggestion, feedback, and improvement to our articles. We have improve several change according to the suggestions and input we received, starting from the introduction, the conceptual framework by adding a definition of community empowerment, improved methodology, research result, and conclusions.

Thank you very much, we hove that the improvement we have met expectation of the reviewer.

Regards,

Author

Reviewer 2 Report

As far as I can tell, this is a very important, new, interesting
and well-conducted study, which I would like to warmly congratulate.

At the same time, I may not be the perfect reviewer, because I know
a lot about urban farming, but I have no expertise in slums in Asia.
So my comments may not be the most profound.
The whole theamtic presented here is extremely complex, as is the
study in its design and methodology. Therefore I actually only have
a few basic questions or comments that I could not recognize when
reading:

1.) Is the collected and evaluated data about urban farming
activities in the described regions that already exist or that
could exist? That is, it was surveyed where there are which urban
farming activities already exist, when they were created, by whom,
how they were established, etc. or only which one could be made
and what effects would that have?

2.) From my understanding of urban farming, such activities have
not only the goal of food production, but also a very large and
important social component. Urban farming initiatives are being
launched in many places to make the region safer (less crime,
vandalism, property destruction), more social cohesion, less
racism, more communication and interaction, more social support
and help among others etc.
If these aspects are not given, in my
opinion it is not urban farming, but private gardening so as not
to starve.
I could not read this aspect out sufficiently in the paper and I
would ask that it be elaborated or incorporated more clearly.

Author Response

Dear Reviewer

We hereby thank you for suggestions, feedback, and improvement to our articles. We have improve several changes according to the suggestions and infut we received, starting from introduction, the conceptual framework by adding a definition of community empowerment, improved methodology, research result, and conclusions.

Thank you very much, we hope that the improvements we have met the expectation of the reviewers.

Regards,

Author

Reviewer 3 Report

Your paper is good but not enough focus on targeted objective. 

(Too) Large approach and the reader cannot appreciate the specific problem to be solve.

In one hand your paper is focus on urban territorial of a slums with good data collection and explanation. But you also consider the specific point of urban farms and practices and the business capacity of some communiity (the level is not clear and the methodology not formulated with precision).

So I recommend to concentrate your paper on a more closed topic with adapted reduction of content. At the same time, give more detailed information on your methodology.

Author Response

Dear Reviewer

We hereby thank you for suggestions, feedback, and improvement to our articles. We have improved several changes according to the suggestions and input we received, starting from the intruduction, the conceptual framework by adding a definition of community empowerment, improved methodology, research result, and conclusions.

Thank you very much, we hope that improvements we have made have met the expectations of the reviewers.

Regards,

Author

Round 2

Reviewer 3 Report

Thank you for the comparative table. Unfortunatelly I don't find adapted change in the new version. You don't upgrade the paper taken into consideration new orientation as suggested. The changes are too superficial.

Author Response

Dear Reviewer

Thank you for your input, suggestions, and criticism of our article.

Here are some of the revision we have made to our article:

  1. The introduction underwent same improvements (pages 3-4)
  2. Conceptual Framework (pages 6-7)
  3. Fundamental changes in the methodology (Pages 7,10,11,12,13 and 14)
  4. Revision of the conclusions (pages 36-37)

This is the information we provide. Once again, thank you for your very valuable suggestions and input to improve our article.

Regards,

Author

Round 3

Reviewer 3 Report

You do your best to improve the paper.